# Impact of acidity and surface modulated acid dissociation on cloud response to organic aerosol

Gargi Sengupta[a], Minjie Zheng[a, b], and Nønne L.  Prisle[a*]

[a]Center for Atmospheric Research, University of Oulu, P.O. Box 4500, 90014 Oulu, Finland
[b]Current affiliation: Institute for Atmospheric and Climate Science, ETH Zurich, Zürich, Switzerland

**Correspondence:** Nønne L. Prisle (Nonne.Prisle@oulu.fi)

**Abstract.** Acid dissociation of the organic aerosol fraction has the potential to impact cloud activating properties by altering aqueous phase $H^+$ concentrations and water activity, but is currently overlooked in most atmospheric aerosol models. We implemented a simple representation of organic acid dissociation in the aerosol-chemistry-climate box model ECHAM6.3-HAM2.3 and investigated the impact on aerosol forming aqueous Sulfur chemistry, cloud droplet number concentrations, and short-wave radiative effect. Many atmospheric organic acids are also surface active and may be strongly adsorbed at the surface of small aqueous droplets. The degree of dissociation has recently been observed for several atmospheric surface-active organics with Brönsted acid character to be significantly shifted in the surface, compared to the bulk aqueous solution. In addition to the well known bulk acidity, we therefore introduced an empirical account of this surface modulated dissociation to further explore the potential impact on aerosol climate effects. Malonic acid and Decanoic acid were used as proxies for atmospheric organic aerosols of different surface-active and acid strengths. Both acids were found to yield sufficient Hydrogen ion concentrations from dissociation in an aqueous droplet population to strongly influence aqueous aerosol Sulfur chemistry, leading to enhanced cloud droplet number concentrations and a cooling short-wave radiative effect. Further considering the surface modulation of organic acid dissociation, the impact on cloud microphysics was smaller than according to the well known bulk solution acidity, but still significant. Our results show that organic aerosol acid dissociation can significantly influence predictions of aerosol and cloud droplet formation and aerosol-cloud-climate effects and that, even for a well known bulk solution phenomenon such as acidity, it may be important to also consider the specific influence of surface effects when surface-active acids comprise a significant fraction of the total organic aerosol mass.

## 1   Introduction

Atmospheric aerosols are an important contributor to Earth's climate. They may either absorb or reflect heat and sunlight, directly affecting Earth's energy budget (Stocker et al., 2014; Masson-Delmotte et al., 2021). Aerosols also contribute to the global climate through indirect effects where they serve as the necessary seeds for cloud formation (Twomey, 1977; Lohmann and Lesins, 2002). The chemical composition of aerosols is complex and includes numerous organic and inorganic species (O'Dowd et al., 2004; Putaud et al., 2010; Murphy et al., 2006). Organic compounds have been reported to comprise approximately $20-50\%$ of the total aerosol mass at mid-latitude regions (Saxena and Hildemann, 1996; Putaud et al., 2004) and much

higher (approximately 90%) in tropical forests (Andreae and Crutzen, 1997; Roberts et al., 2001; Kanakidou et al., 2005). Significant amounts of organic aerosols (approximately 70% of the total aerosol mass) are also reported in the middle troposphere (Huebert et al., 2004). Despite their abundance and importance, the organic fraction is the least understood component of atmospheric aerosols and the uncertainty around organic aerosols and their interaction with clouds remains one of the largest overall sources of uncertainty in climate projections (IPCC, 2013; Seinfeld et al., 2016; Legg, 2021). The size, chemical composition, and phase-state of organic aerosol (OA) are known to directly impact its cloud droplet formation potential (Hallquist et al., 2009; McFiggans et al., 2006) and the radiative effect of clouds (Turnock et al., 2019). However, climate models often have a limited representation of OA processes and properties, such as acidity and surface activity of the organic components (Kanakidou et al., 2005; Prisle et al., 2012a; Freedman et al., 2018; Pye et al., 2020), which contribute to the overall uncertainty in predictions of OA and their interactions with clouds.

Organic aerosols contain a substantial fraction of species exhibiting Brönsted acid character (Jacob, 1986; Millet et al., 2015; Keene and Galloway, 1984; Chebbi and Carlier, 1996; Chen et al., 2021b; Angelis et al., 2012; Mochizuki et al., 2016; Wu et al., 2020; Kawamura et al., 1985). The concentrations of acidic species in aqueous aerosols directly affect the aerosol pH by modifying the $H^+$ concentrations within the aerosol (Pye et al., 2020; Ault, 2020). This pH affects the dissociation of individual acidic species with significant consequences for aerosol chemistry (Hung et al., 2018; Wang et al., 2018) and phase state (Liu et al., 2019). For example, pH dependent Sulfur oxidation (Liu et al., 2020) and salt formation by acidic or basic OA (Yli-Juuti et al., 2013) can each lead to significant mass formation and alter the overall chemical composition of aerosols. The chemical form (protonated or deprotonated) of acidic OA and contributions to the number of solute species in the aqueous aerosol phase can strongly affect water activity and condensation–evaporation equilibrium (Prisle, 2006; Prisle et al., 2008; Frosch et al., 2011; Michailoudi et al., 2019).

Many atmospheric organic acids also exhibit surface activity in aqueous solutions, such as aqueous aerosols and cloud droplets (Prisle, 2023). Surface active organics (surfactants) have been reported in atmospheric aerosols from many different regions and environments (Gérard et al., 2016; Petters and Petters, 2016; Nozière et al., 2017; Kroflič et al., 2018; Gérard et al., 2019a). Surfactants adsorb at the aqueous surface, leading to enhanced surface concentrations compared to the interior (bulk) of a solution. In microscopic and submicron-sized aerosols and droplets, the surface adsorption can result in significant redistribution of surface active OA mass from the bulk to the surface phase, so-called bulk–surface partitioning, as a consequence of the very high surface area ($A$) to bulk volume ($V$) ratio in these size ranges (Prisle et al., 2008, 2010b). For spherical droplets of diameter $D_{wet} = 0.1$, 1, and 10 μm, $A/V = 6/D_{wet}$ is 60, 6, and 0.6 μm$^{-1}$, respectively (Prisle, 2021). Thermodynamic calculations have shown that for aerosol particles containing surfactant fatty acids and their salts, organosulfates, di- and polycarboxylic acids, and complex fulvic acids, a large fraction of the surface active OA is partitioned to the surface during major parts of hygroscopic growth and cloud droplet activation (Prisle et al., 2010b, 2011; Hansen et al., 2015; Malila and Prisle, 2018; Lin et al., 2018, 2020; Prisle, 2021; Vepsäläinen et al., 2022, 2023). Consequently, the chemical and physical state of the surface may significantly contribute to determining the overall aerosol properties (Prisle et al., 2012b; Bzdek et al., 2020; Prisle, 2021, 2023).

Highly surface sensitive synchrotron radiation excited X-ray photoelectron spectroscopy (XPS) measurements have been used to investigate the acid–base speciation of surface-active atmospheric Brönsted acids and bases in the surface region of aqueous solutions (Prisle et al., 2012b; Öhrwall et al., 2015b; Werner et al., 2018). In these experiments, the protonated form of each conjugate pair was found at an extraordinarily large fraction, compared to that expected from their acidity and the bulk solution pH. Werner et al. (2018) observed a shift in the degree of protonation for simple mono-carboxylic acids at the surface in dilute aqueous solutions (50 mM Butyric and Pentanoic acid) corresponding to activating cloud droplets. The acid–base equilibrium in the surface was shifted systematically across a very wide range of solution pH, overall corresponding to an apparent shift in $pK_a$ on the order of $1-2$ pH units, compared to the well known bulk acidity, for each of the acids. Shifts in surface protonation degree of similar magnitude were also previously observed using XPS for dilute aqueous solutions of $10-25$ mM Decanoate/Decanoic acid (Prisle et al., 2012b), 0.1 M Propanoate/Propanoic acid, and 0.1 M Octanoate/Octanoic acid (Öhrwall et al., 2015a) at near-neutral pH. XPS measurements on aqueous solutions of Succinic acid, a moderately surface-active dicarboxylic acid, over a range of concentrations from $0.05-0.5$ M and solution pH from $2.0-12.9$ also indicated a shifted acid–base equilibrium in the surface compared to the bulk, where the protonated form showed a considerably higher propensity to reside in the aqueous surface region than its conjugate deprotonated form (Werner et al., 2014a). These observations are further supported by experiments by Wellen et al. (2017), who used surface tension titration and infrared reflection absorption spectroscopy to obtain pH dependent aqueous surface tension of Nonanoic and Decanoic acids at their aqueous solution surfaces. They inferred that the so-called surface $pK_a$ was greater than the well known bulk $pK_a$ by 1 pH unit for Nonanoic acid and 2 pH units for Decanoic acid, suggesting that the organic acid dissociation response to a given aqueous bulk pH is different in the surface, compared to the bulk.

The general behavior of acidic compounds at the aqueous interface is still not well constrained (Saykally, 2013). Petersen and Saykally (2005, 2008) observed an enhanced surface concentration of hydronium ions in aqueous solutions of Hydroiodic acid, alkali iodides and alkali hydroxides using second harmonic generation spectroscopy experiments. This was in contrast to previous macroscopic bubble and droplet experiments, which were interpreted to indicate that hydroxide ions were enhanced at the air–water interface (Graciaa et al., 1995; Takahashi, 2005; Karraker and Radke, 2002; Creux et al., 2007). Enami et al. (2010) also made similar observations of enhanced hydronium ions in the surface of Trimethylamine solutions using electrospray mass spectrometry. Recently, Gong et al. (2023) used stimulated Raman scattering microscopy to observe enhanced concentrations of Sulfate and Bisulfate anions, with Bisulfate being more surface enriched than Sulfate, in the surface of $2.9$ μm aerosol droplets generated from an aqueous solution with 300 mM $NaHSO_4$ and 50 mM $Na_2SO_4$ at the same pH. They interpret this as an enhancement of acidity, with approximately threefold increase in the Hydrogen ion concentration, at the droplet edge, compared to the center of the droplet. Previous observations by Margarella et al. (2013) on the dissociation of Sulfuric acid at the water interface using liquid-jet photoelectron spectroscopy, have also reported that the ratio of Bisulfate-to-Sulfate anions was higher in the surface region.

In this work, we use an aerosol–chemistry-climate box model to investigate the potential impact on aerosol forming aqueous phase Sulfur chemistry, cloud droplet activation, and aerosol-cloud-climate parameters of organic acid dissociation in aqueous aerosols and its additional surface modulation for surfactant acidic OA. Very few studies have previously addressed the disso-

ciation of organic components in aerosols in relation to cloud chemistry and microphysics (Tilgner et al., 2021a; Angle et al., 2021). Tilgner et al. (2021a) compiled a kinetic data set to study the implications of varying aerosol acidity on the oxidation of the protonated and deprotonated forms of atmospheric organic acids with aqueous-phase oxidants, such as OH radical, $NO_3$ radical, or $O_3$. They showed that acidity strongly affects the chemical processing of dissociating organic compounds, but did not provide a direct correlation between organic dissociation and cloud activation. Angle et al. (2021) measured the pH of nascent seaspray aerosol using a Micro-Orifice Uniform Deposit Impactor (MOUDI) and impacting the aerosols onto colorimetric pH strips. They found that the pH of freshly emitted (nascent) seaspray aerosols was approximately four pH units lower than that of sea water. The dissociation of organic acids in the aerosols is proposed as a possible factor contributing to the low nascent seaspray aerosol pH. They note that for a nascent seaspray aerosol with a diameter of 200 nm and surface layer of Palmitic acid, only 4.4% acid dissociation would be required to lower the aerosol pH from 8 to 2. However, they do not provide any details on how the organic acid dissociation would affect the aerosol properties. To the best of our knowledge, the organic acid dissociation in aqueous aerosols has never been studied in the context of a cloud activation model, let alone accounting for surface specific modulation of organic acid dissociation in aqueous aerosols.

## 2  Methods

We first introduce an account of well known organic acid dissociation in bulk aqueous solution and then augment it with a simple empirical representation to further include surface-driven suppression of dissociation according to observations from XPS measurements. We use ECHAM6.3–HAM2.3 (here referred to as HAMBOX), which is a box model version of the aerosol–chemistry–climate model ECHAM–HAMMOZ (Tegen et al., 2019), to calculate the total aerosol population Sulfate mass and cloud droplet number concentrations (CDNC) for an air parcel and the short-wave radiative effect (RE) from cloud formation, as examples of key processes taking place in aqueous organic aerosols and droplets. The impact of organic aerosol bulk acidity and surface-modulated suppressed dissociation on aerosol Sulfur chemistry and cloud microphysics is assessed by comparing to predictions for identical conditions without accounting for organic acid dissociation. The simulation time for all calculations was 1 hour, with 1 second time steps.

### 2.1  Aerosol module in HAMBOX

HAMBOX uses the SALSA2.0 aerosol module (Kokkola et al., 2018), where the aerosol size distribution is calculated using the sectional approach (Jacobson, 2005) and represented using 10 size bins $i$. Here, we group the 10 size bins into four sub-ranges: Nucleation ($i = 1$ and 2) with mean particle diameter, $\bar{d}_p = 56$ nm; Aitken ($i = 3$, 4 and 5) with $\bar{d}_p = 160$ nm; Accumulation ($i = 6, 7$ and 8) with $\bar{d}_p = 485$ nm, and Coarse ($i = 9$ and 10) with $\bar{d}_p = 1.85$ µm. The initial number concentration in each sub-range (Table 1) used for all HAMBOX simulations is representative of clean environments, such as European villages (Tunved et al., 2005, 2008). As a property of the sectional approach, when particles grow or shrink out of the boundaries of their size bins, they are redistributed to new size bins and the new aerosol size distribution is calculated at each simulation time step.

**Table 1.** Initial aerosol number concentration in each size sub-range used in HAMBOX–SALSA2.0, representative of clean environments Tunved et al. (2005, 2008).

| Sub-range | Number concentration [$\mathrm{N\,cm^{-3}}$] | Geometric mean diameter [µm] & Standard deviation |
|---|---|---|
| Nucleation | 100 | 0.01 & 1.50 |
| Aitken | 400 | 0.3 & 1.50 |
| Accumulation | 200 | 1.0 & 1.50 |
| Coarse | 0 | 3.0 & 2.0 |

A Sulfur chemistry module (Feichter et al., 1996a) is coupled to the aerosol growth module in SALSA2.0. At each time step, the Sulfur chemistry module feeds the calculated Sulfate mass fraction in the aerosol population into the growth module, which undergoes an aerosol redistribution and feeds back the new aerosol size distribution and chemical composition to the Sulfur chemistry module. The aerosol chemical composition in SALSA2.0 is represented by model compound classes 'Sulfate' (SU), 'Organic aerosol' (OA), 'Sea salt' (SS), 'Black carbon' (BC), and 'Mineral dust' (DU). Of these model compounds,

Sulfate, Organic aerosol, and Sea salt constitute the soluble species and are considered as internally mixed in each size bin of the aerosol population. Black carbon and Mineral dust are insoluble species which are externally mixed in each size bin with the soluble species, as described by Kokkola et al. (2018). We consider five different initial conditions with different OA mass fractions $\chi_{OA} = \{0.2, 0.4, 0.6, 0.8, 1\}$ to represent different environments where OA have been reported in varying concentrations. For example, in boreal forest environments, OA mass fraction is reported around $0.6$ (Äijälä et al., 2019), and in

marine environments, around $0.2$ (O'Dowd et al., 2004). $\chi_{OA} = 1$ is a hypothetical extreme where we consider OA to comprise the entire aerosol phase. The initial mass fractions of all aerosol model compounds in these five conditions are shown in Table 2.

     The Sulfur chemistry module is used to calculate the aqueous phase secondary Sulfate mass from oxidation of Sulfur dioxide in the aerosol population based on varying Hydrogen ion concentration in the aqueous aerosol, considering no organic acid

dissociation, organic acid dissociation according to well known bulk acidity, and surface modulated organic acid dissociation (Section 2.3 below).

**Table 2.** Initial aerosol mass fractions of all model compounds in the five different environmental scenarios considered.

| Organic ($\chi_{OA}$) | Sulfate ($\chi_{SU}$) | Black carbon ($\chi_{BC}$) | Sea salt ($\chi_{SS}$) | Mineral dust ($\chi_{DU}$) |
|---|---|---|---|---|
| 0.2 | 0.4 | 0.05 | 0.1 | 0.25 |
| 0.4 | 0.4 | 0.05 | 0.05 | 0.1 |
| 0.6 | 0.4 | 0 | 0 | 0 |
| 0.8 | 0.2 | 0 | 0 | 0 |
| 1 | 0 | 0 | 0 | 0 |

## 2.2 Cloud microphysics in HAMBOX

From the total aerosol mass and composition, the resulting cloud droplet number concentrations and consequent short-wave radiative effect are calculated with the HAMBOX cloud microphysics module. The HAMBOX cloud microphysics used in this work includes the calculation of critical supersaturation ($S_i$) and activated fraction ($n_i$) for each aerosol size bin $i$. A detailed description of the parameterizations and equations used to calculate these cloud activation factors are available from Abdul-Razzak (2002) and Abdul-Razzak et al. (1998) and briefly summarized here.

First, the maximum critical supersaturation for the air parcel is calculated as

$$S_{\mathrm{max}} = \frac{S_e}{\left[ 0.5 \left( \frac{\varsigma}{\eta} \right)^{3/2} + \left( \frac{S_e^2}{\eta+3\zeta} \right)^{3/4} \right]^{1/2}}, \tag{1}$$

where $\eta$ is the surface tension correction factor, $\zeta$ is the correction factor for the Kelvin term in the Köhler curve (for details, see eqs. 5 and 6 in Abdul-Razzak and Ghan (2002)), and $S_e$ is the effective critical supersaturation for the air parcel,

$$S_e^{2/3} = \frac{\sum_{i=1}^{I} N_i}{\sum_{i=1}^{I} N_i / S_i^{2/3}}. \tag{2}$$

Here, $I = 10$ is the total number of aerosol size bins, $N_i$ is the number of particles in each bin $i$, and $S_i$ is the critical supersaturation for each bin, given by

$$S_i = \exp \left( \frac{A}{D_{\mathrm{wet}}} - \frac{B}{D_{\mathrm{wet}}^3 - d_p^3} \right) - 1, \tag{3}$$

where $D_{\mathrm{wet}}$ and $d_p$ are droplet and dry particle diameters, respectively. The terms $A$ and $B$ are calculated as

$$A = \frac{4 M_w \sigma_w}{R T \rho_w}, \quad B = \frac{6 n_s M_w}{\pi \rho_w}, \tag{4}$$

where $M_w = 0.018 \ \mathrm{kg \, mol^{-1}}$ is the molecular weight of water, $\sigma_w = 0.073 \ \mathrm{N \, m^{-1}}$ is the surface tension of pure water, $\rho_w = 1000 \ \mathrm{kg \, m^{-3}}$ is the density of water, $R = 8.314 \ \mathrm{J \, K^{-1} mol^{-1}}$ is the ideal gas constant, $T = 293 \ \mathrm{K}$ is the temperature, and $n_s$ is the number of moles of solute obtained from the mass fractions of soluble species $\chi_{\mathrm{OA}}$, $\chi_{\mathrm{SU}}$, and $\chi_{\mathrm{SS}}$.

With the maximum supersaturation of the air parcel, cloud droplet activation in each bin is determined by comparing $S_{\mathrm{max}}$ with $S_{il}$ and $S_{iu}$ (the lower and upper critical supersaturation bounds of the bin). The number of activated particles in each size bin $i$ is given by

$$n_i = 0, \quad \text{if} \quad S_{\mathrm{max}} < S_{il}, \tag{5}$$

$$n_i = \frac{\log(S_{\mathrm{max}}/S_{il})}{\log(S_{iu}/S_{il})}, \quad \text{if} \quad S_{il} \leq S_{\mathrm{max}} \leq S_{iu}, \tag{6}$$

and

$$n_i = 1, \quad \text{if} \quad S_{iu} < S_{\mathrm{max}}. \tag{7}$$

$S_{il}$ and $S_{iu}$ are obtained using eq. 3 for the diameters of the smallest ($d_{il}$) and largest ($d_{iu}$) particles in each size bin. The average activated fraction for all size bins is then calculated by

$$n = \frac{\sum n_i}{\sum N_i}. \tag{8}$$

The total number of activated particles is given by the cloud droplet number concentration, which is calculated using the number of activating particles within each size bin, as

$$\text{CDNC} = \sum_{i=1}^{I} N_i n_i. \tag{9}$$

The CDNC calculated using eq. 9 considering organic bulk acidity and surface modulated organic acid dissociation is denoted by $\text{CDNC}_{\text{HA}}$ and the change with respect to the reference condition of no organic acid dissociation ($\text{CDNC}_0$) is

$$\Delta\text{CDNC} = \frac{\text{CDNC}_{\text{HA}} - \text{CDNC}_0}{\text{CDNC}_0}. \tag{10}$$

We estimate the change in short-wave radiative effect (RE) from including organic bulk acidity and surface modulated organic acid dissociation, respectively, using the method given by Bzdek et al. (2020). The change in cloud-top albedo at constant cloud liquid water content ($\text{LWC} = 0.03\,\text{g}\,\text{m}^{-3}$, Thompson (2007)) is calculated from $\Delta\text{CDNC}$ as

$$\Delta a = \text{LWC}(1 - \text{LWC})\Delta\text{CDNC}/(3\,\text{CDNC}). \tag{11}$$

The short-wave radiative effect is then calculated as

$$\text{RE} \approx -\text{F}_0\text{E}_{\text{LWC}}\text{T}_{\text{LWC}}^2\Delta a, \tag{12}$$

where $F_0 = 340\,\text{W}\,\text{m}^{-2}$ is the incoming solar flux at the top of the atmosphere, $E_{\text{LWC}} = 0.3$ is the fractional coverage of different types of clouds, and $T_{\text{LWC}} = 0.76$ is the transmittance of the atmosphere at visible wavelengths, which is assumed to be constant for all simulations.

For all HAMBOX cloud microphysics calculations, we assume a constant cloud temperature of 271 K, cloud pressure of 101 kPa, cloud fraction of 0.3, saturation ratio of gas phase water of 0.3, and updraft velocity of $0.3\,\text{m}\,\text{s}^{-1}$, consistent with Tegen et al. (2019).

## 2.3 Sulfur chemistry in HAMBOX

We use the aqueous Sulfur chemistry module of Feichter et al. (1996b), with modifications described below, to calculate the aqueous phase secondary Sulfate concentration $[\text{SO}_4^{2-}]''$ in the aerosol population formed from the oxidation of $\text{SO}_2$ by $\text{H}_2\text{O}_2$ and $\text{O}_3$ in aqueous droplets. The reaction rate for the $\text{H}_2\text{O}_2$ oxidation pathway can be written as:

$$\frac{\partial}{\partial t}\left[\text{SO}_4^{2-}\right]'' = \frac{k_4\left[\text{H}_2\text{O}_2\right]\left[\text{SO}_2\right]}{\left[\text{H}^+\right] + 0.1} \tag{13}$$

where the rate constant $k_4$ is calculated by

$$k_4 = 8 \times 10^4 \exp\left(-3650\left(\frac{1}{T} - \frac{1}{298}\right)\right), \tag{14}$$

where $T$ is the cloud temperature = 271 K. Equation 13 is known to be pH insensitive (Liu et al., 2020) and is used in this work to determine the aqueous secondary Sulfate concentration from the $H_2O_2$ oxidation for simulations where organic acid dissociation is not considered.

To calculate $[SO_4^{2-}]''$ from the $H_2O_2$ oxidation pathway accounting for pH dependency arising from organic bulk acidity, we follow the procedure given by Liu et al. (2020), which is valid for pH > 2. Here, we use the general acid catalysis reaction mechanism, where $SO_2$ in an aqueous environment exists as the $HSO_3^-$ anion and reacts with $H_2O_2$ in the presence of an organic acid (HA) catalyst, which acts as a proton donor (Maaß et al., 1999; McArdle and Hoffmann, 1983). Briefly, the overall reaction mechanism is represented as

$$HSO_3^- + H_2O_2 \rightleftharpoons HOOSO_2^- + H_2O \qquad (R1)$$

and

$$HOOSO_2^- + HA \rightarrow 2H^+ + SO_4^{2-} + A^-. \qquad (R2)$$

The rate expression for R1–R2 is

$$\frac{\partial}{\partial t}\left[SO_4^{2-}\right]'' = \left(k + \frac{k_{HA}[HA]}{[H^+]}\right) K_{a1}[SO_2][H_2O_2], \qquad (15)$$

where $K_{a1}$ is the thermodynamic dissociation constant of $H_2SO_3$ and $k$ is a constant derived from the reaction rate coefficient and the thermodynamic equilibrium constants. $k_{HA}$ is the overall rate constant for the general acid catalysis mechanism approximated by $\log k_{HA} = -0.57(pK_a) + 6.83$ (Liu et al., 2020; Drexler et al., 1991). This approximation for $k_{HA}$ in relation to the $pK_a$ of an organic acid was derived by Liu et al. (2020) for an ionic strength of $I = 0.5\,\mathrm{mol\,kg^{-1}}$. Therefore, we assume this same ionic strength for aqueous droplets in all our calculations.

The secondary Sulfate concentration from $O_3$ oxidation is given by

$$\frac{\partial}{\partial t}\left[SO_4^{2-}\right]'' = \left(k_{51} + \frac{k_{52}}{[H^+]}\right)[O_3][SO_2], \qquad (16)$$

where rate constants $k_{51}$ and $k_{52}$ are calculated from

$$k_{51} = 4.39 \times 10^{11} \exp\left(\frac{-4131}{T}\right) \qquad (17)$$

and

$$k_{52} = 2.56 \times 10^3 \exp\left(\frac{-996}{T}\right). \qquad (18)$$

    Sulfate concentrations thus calculated in the aqueous phase Sulfur chemistry module is distributed to pre-existing aerosol size bins in SALSA2.0.

    When no organic aerosol acid dissociation (no diss) is considered, the default $H^+$ concentration in HAMBOX is denoted by $[H^+]_0$ and obtained from water and aqueous phase Sulfate concentrations as

$$\left[H^+\right]_0 = \left[H^+\right]_{initial} + \frac{[SO_4^{2-}]_{sol}}{LWC \times MW_{SO_4^{2-}}}, \qquad (19)$$

where LWC is the cloud liquid water content in $[\mathrm{g\,m^{-3}}]$, $\mathrm{MW_{SO_4^{2-}}}$ is the molar weight of the Sulfate anion $(\mathrm{g\,mol^{-1}})$, and the soluble Sulfate concentration $[\mathrm{SO_4^{2-}}]_{\mathrm{sol}}$ is obtained from the summation of soluble Sulfate (obtained from $\chi_{\mathrm{SU}}$, Table 2) in all bins. $[\mathrm{H^+}]_{\mathrm{initial}} = 2.5 \times 10^{-6}\ \mathrm{mol\,L^{-1}}$ is the Hydrogen ion concentration obtained from the cloud pH = 5, which is assumed to be uniform for all size bins and consistent with the pH of warm low lying tropospheric clouds (Pye et al., 2020). For all simulations in the Sulfur chemistry module, we assume $[\mathrm{SO_2}]$, $[\mathrm{H_2O_2}]$, and $[\mathrm{O_3}]$ in cloud are fixed at 5 ppb, 1 ppb, and 50 ppb, respectively (Tilgner et al. (2021b)).

We introduce organic acid dissociation to the Sulfur chemistry module by modifying eq. 19 to obtain the total Hydrogen ion concentration in the aerosol population as

$$\left[\mathrm{H^+}\right]_{\mathrm{tot}} = \left[\mathrm{H^+}\right]_{\mathrm{initial}} + \frac{\left[\mathrm{SO_4^{2-}}\right]_{\mathrm{sol}}}{\mathrm{LWC} \times \mathrm{MW_{SO_4^{2-}}}} + \left[\mathrm{H^+}\right]_{\mathrm{HA}}, \tag{20}$$

where $[\mathrm{H^+}]_{\mathrm{HA}}$ is the concentration of the Hydrogen ions dissociated by the acidic organic aerosol components. The calculated $[\mathrm{H^+}]_{\mathrm{tot}}$ is then used to obtain the aqueous phase secondary Sulfate concentration in the aerosol population from $\mathrm{SO_2}$ oxidation by $\mathrm{H_2O_2}$ (eq. 15) and $\mathrm{O_3}$ (eq. 16), for varying conditions of organic acid dissociation. The Sulfate concentrations thus obtained gives a modified Sulfate mass fraction ($\chi_{\mathrm{SU}}$, Section 2.1 and Table 2) in the entire aerosol population.

## 2.4 Organic acid dissociation

We assume the entire OA fraction is comprised of an organic acid and consider two different acids, Malonic acid (a diprotic acid) and Decanoic acid (a monoprotic acid), as examples of important organic aerosol components in the atmosphere (Yassaa et al., 2001; Narukawa et al., 2002; Mochida et al., 2003; Graham et al., 2003; Cheng et al., 2004; Li and Yu, 2005; Tedetti et al., 2006) with different well known aqueous bulk acidity and prominent examples of moderately and strongly surface active organic species, respectively (Vepsäläinen et al., 2022, 2023). The acid dissociation constants for aqueous bulk solutions are here denoted as $pK_a^{\mathrm{bulk}}$ to distinguish the well known bulk dissociation behavior from surface modulated dissociation introduced in Section 2.4.3. The molecular weight (MW), density ($\rho$), and bulk acid constants for Decanoic and Malonic acids used in our calculations are given in Table 3. For the monoprotic Decanoic acid, $pK_a^{\mathrm{bulk}}$ is the reported first $pK_a$ readily available from literature, whereas for the diprotic Malonic acid, $pK_a^{\mathrm{bulk}}$ is taken as the sum of the first and second $pK_a$ reported in literature (same as $\beta$ in eq. 29 below). The dissociation behavior of monoprotic and diprotic acids in similar aqueous environments differ greatly and we use different kinetic equations to describe both treatments of organic bulk and surface modulated acid dissociation, presented in the following Sections 2.4.1, 2.4.2, and 2.4.3.

### 2.4.1 Monoprotic acids

The dissociation of a monoprotic organic acid (HA) in aqueous solution is represented by the equilibrium

$$\mathrm{HA + H_2O \rightleftharpoons A^- + H_3O^+}, \tag{R3}$$

where $\mathrm{H^+}$ from the dissociation of the organic acid are considered as fully hydrated, such that the concentration of $\mathrm{H_3O^+}$ is equivalent to the concentration of Hydrogen ions from dissociation of HA.

The equilibrium acid dissociation constant is

$$K_a = \frac{a_{H_3O^+} a_{A^-}}{a_{HA}}, \tag{21}$$

where $a_{H_3O^+}, a_{A^-}$ and $a_{HA}$ are the activities of $H_3O^+$ cations, $A^-$ anions and HA molecules, respectively. Equation 21, can be approximated in terms of the molar concentrations and ideal-dilute molar concentration based activity coefficients ($\gamma_i$) of each species $i$ as

$$K_a = \frac{[H_3O^+][A^-]}{[HA]} \frac{\gamma_{H_3O^+} \gamma_{A^-}}{\gamma_{HA}}. \tag{22}$$

Since a monoprotic acid HA has only one ionizable Hydrogen, in eq. 22

$$[A^-] = [H_3O^+], \text{ and } [HA] = [HA]_{tot} - [H_3O^+], \tag{23}$$

where $[HA]_{tot}$ is the total concentration of the organic acid. The acid dissociation degree $\alpha$ is defined as

$$\alpha = \frac{[A^-]}{[HA]_{tot}} = \frac{[H_3O^+]}{[HA]_{tot}}. \tag{24}$$

Combining eq. 22 and 24 and approximating $\frac{\gamma_{H_3O^+} \gamma_{A^-}}{\gamma_{HA}}$ with the mean activity coefficient $\gamma_{\pm}$,

$$K_a = [HA]_{tot} \left( \frac{\alpha^2}{1 - \alpha} \right) \gamma_{\pm}^2. \tag{25}$$

For a highly dilute solution (e.g., $[HA]_{tot} < 0.001 \text{ mol L}^{-1}$), $\gamma_{\pm}^2 \approx 1$ and eq. 25 can be written as

$$\alpha = \frac{-K_a + \sqrt{K_a^2 + 4K_a \times [HA]_{tot}}}{2[HA]_{tot}}. \tag{26}$$

With $[HA]_{tot}$ known from $\chi_{OA}$ and other properties of the aerosol population, the Hydrogen ion concentration from organic acid dissociation $[H^+]_{HA}$ is obtained as $[H_3O^+]$ in eq. 24 by

$$[H^+]_{HA} = [H_3O^+] = \alpha[HA]_{tot}, \tag{27}$$

where the acid dissociation degree $\alpha$ is given by eq. 26.

### 2.4.2 Diprotic acids

For a diprotic organic acid ($H_2A$), the dissociation of $H^+$ ions in an aqueous solution can be considered to occur in two stages,

$$H_2A + H_2O \rightleftharpoons HA^- + H_3O^+ \tag{R4}$$

and

$$HA^- + H_2O \rightleftharpoons A^{2-} + H_3O^+. \tag{R5}$$

The dissociation constant for R4 is the first dissociation constant of the diprotic acid, denoted as $K_{a1}$, and the dissociation constant for R5 is the second dissociation constant of the diprotic acid, denoted as $K_{a2}$. The overall dissociation constant of $H_2A$ is

$$\beta = K_{a1}K_{a2}, \tag{28}$$

and therefore,

$$p\beta = pK_{a1} + pK_{a2}. \tag{29}$$

Using similar assumptions as for the monoprotic acid, for a highly dilute solution, the acid dissociation degree $\alpha$ for a diprotic acid can be derived as

$$\alpha = \frac{1}{4\beta[H_2A]_{tot} + 2}, \tag{30}$$

where $[H_2A]_{tot}$ is the total concentration of the diprotic acid. Analogously to the case of a monoprotic acid, the Hydrogen ion concentration from dissociation of a diprotic organic acid $[H^+]_{HA}$ is given by

$$[H^+]_{HA} = [H_3O^+] = \alpha[H_2A]_{tot}, \tag{31}$$

where the acid dissociation degree $\alpha$ is now given by eq. 30.

### 2.4.3 Surface modulated organic acid dissociation

We now introduce a simple empirical representation of the shift in organic acid dissociation previously observed in surface-sensitive XPS experiments. Werner et al. (2018) found that the surface specific dissociation state of surface active mono-carboxylic acids was significantly suppressed in dilute aqueous solutions across a very wide range of solution $pH = 2 - 12$. Similar suppressed dissociation states were also found for other mono- and dicarboxylic acids of both stronger and weaker surface activity, in aqueous solutions closer to neutral pH (Prisle et al., 2012b; Werner et al., 2014b; Öhrwall et al., 2015a). The shifted dissociation states are attributed to both increased concentrations of the surface active organic acids in the surface and increased non-ideality (higher activity coefficients) of the charged deprotonated conjugate species $A^-$ and hydronium ions, compared to the neutral molecular acid HA, in the organic-rich air–solution interfacial region (Werner et al., 2018; Prisle, 2023). From eq. 22, this corresponds to an apparent shift of the acid $pK_a$ at the surface,

$$pK_a = pK_a^{bulk} + \log\left(\frac{\gamma_{H_3O^+}\gamma_{A^-}}{\gamma_{HA}}\right), \tag{32}$$

compared to the well known bulk acidity $pK_a^{bulk}$ obtained for dilute aqueous solutions, where all activity coefficients are assumed to be ideal, $\gamma_i = 1$ (Prisle, 2023).

The dissociation states observed with XPS are broadly consistent with a magnitude of the apparent shift in $pK_a$ of $\log\left(\frac{\gamma_{H_3O^+}\gamma_{A^-}}{\gamma_{HA}}\right) = 1 - 2$ pH units across the surface titration curve (Prisle, 2023). We here introduce the effect of surface modulated acid dissociation by shifting the well known bulk $pK_a$ of each organic acid according to these shifts of the surface titration curves. We

consider two magnitudes of this apparent shift, covering the range of experimental observations from XPS. For a monoprotic acid, we consider $pK_a = pK_a^{\mathrm{bulk}} + 1$ and $pK_a = pK_a^{\mathrm{bulk}} + 2$, where $pK_a^{\mathrm{bulk}}$ is the well known $pK_a$ of the organic acid in aqueous bulk solution. To represent the surface shifted dissociation of both carboxylic groups in a diprotic acid, we increase both the first and second acid constant, by 1 or 2 pH units, to similarly obtain $pK_a^{\mathrm{bulk}} + 1$ and $pK_a^{\mathrm{bulk}} + 2$. We here refer to the shifted $pK_a$ values as the surface modulated *apparent* $pK_a$. However, we strongly emphasize that the $pK_a$, which is an intrinsic property of each organic acid in bulk aqueous solution, is not itself changed. Only the dissociation responses of the organic acids to a given pH of the solution (here, the cloud pH) are changed in the surface (Prisle, 2023).

For both mono- and diprotic acids, the values used for surface modulated apparent $pK_a$ are given in Table 3. For each $pK_a$, the corresponding acid dissociation degree $\alpha$ is calculated for the monoprotic acid using eq. 26 and for the diprotic acid using eq. 30. The value for $\alpha$ decreases with increasing $pK_a$, such that the increased apparent $pK_a$ represent suppressed dissociation of the organic acid in the surface. The surface modulation of organic dissociation is most pronounced in a range of several pH units around the bulk $pK_a$. At very low and very high pH, the surface dissociation states collapse onto the well known bulk solution dissociation behavior (Werner et al., 2018; Prisle, 2023). For both the organic acids used here, the $pK_a^{\mathrm{bulk}}$ is within a few pH units of the cloud pH.

Although we implement the effect of surface modulated acid dissociation as a consequence of simultaneous surface activity of the organic acid, we do not explicitly consider the bulk–surface partitioning of organic acids in our calculations. Our simple empirical representation by shifting the apparent $pK_a$ for the organic acid corresponds to assuming that the overall dissociation state in aqueous aerosols and droplets is described by the surface modulated properties. This is closely representative of aerosols and droplets where the majority of organic aerosol components are partitioned to the aqueous surface, as a consequence of strong organic surface activity or high $A/V$ in the microscopic and submicron size ranges (Prisle, 2021, 2023). In real atmospheric aerosol and droplet mixtures of both surface active and more water soluble OA, organic species will be partially partitioned to the surface and the overall dissociation state should be described as a combination of both well known bulk acidity and surface modulated states. The present simple empirical representation therefore gives an upper bound of the potential effects of surface modulated acid dissociation according to the previous observations from XPS experiments.

When surface modulated organic acid dissociation is considered, these properties are assumed to remain consistent throughout the 1-hour simulations. Prisle et al. (2008) and Prisle (2021) estimated that surface adsorption of typical atmospheric surfactants equilibrate within a timescale of a second in micron-sized droplets. Lin et al. (2020) investigated the impact of surface adsorption dynamics on surfactant effects in cloud droplet activation and found that different dynamic effects nearly cancel out at every time step. Noziere et al. (2014) assumed that both the bulk and surface reach a state of reasonable equilibrium with respect to organic adsorption at the aqueous surface within approximately 495 seconds. Therefore, we consider this assumption to be a reasonable first approximation.

**Table 3.** Properties of the organic acids used in calculations of acid dissociation, including molecular properties (MW and $\rho$), well known bulk solution acidity ($pK_{a1}$, $pK_{a2}$, and $pK_a^{\text{bulk}}$), and surface modulated dissociation properties (implemented as $pK_a^{\text{bulk}}+1$ and $pK_a^{\text{bulk}}+2$).

|  | Malonic acid | Decanoic acid |
|---|---|---|
| Molecular weight, MW | $104 \text{ g mol}^{-1}$ | $172.26 \text{ g mol}^{-1}$ |
| Density, $\rho$ | $1.62 \text{ g cm}^{-3}$ | $0.893 \text{ g cm}^{-3}$ |
| $pK_{a1}$ | $2.8^a$ | $4.9^b$ |
| $pK_{a2}$ | $5.7^a$ | - |
| $pK_a^{\text{bulk}}$ | 8.5 | 4.9 |
| $pK_a^{\text{bulk}}+1$ | 10.5 | 5.9 |
| $pK_a^{\text{bulk}}+2$ | 12.5 | 6.9 |

$^a$ Stahl and Wermuth (2002). $^b$ Martell and Smith (1974).

### 2.4.4 The van't Hoff factor for organic dissociation

Aqueous phase dissociation also influences the available amount of solute species in aerosol particles and droplets, affecting the calculations of water activity and critical supersaturation (Section 2.2). The molar amount of available solute $n_s$ is calculated in SALSA2.0 from the the molar amounts of all the internally mixed soluble species as

$$n_s = i_{\text{SU}} n_{\text{SU}} + i_{\text{OA}} n_{\text{OA}} + i_{\text{SS}} n_{\text{SS}}, \tag{33}$$

where the $n_{\text{SU}}$, $n_{\text{OA}}$ and $n_{\text{SS}}$ are the molar amounts of Sulfate, Organic aerosol, and Sea salt, respectively, derived from the initial aerosol mass fractions given in Table 2 and $i_{\text{SU}}$, $i_{\text{OA}}$ and $i_{\text{SS}}$ are the corresponding van't Hoff factors for each soluble species. In SALSA2.0, the Sulfate and Sea salt are considered as fully dissociated, such that $i_{\text{SU}} = 3$ and $i_{\text{SS}} = 2$. By default, Organic aerosol is not considered as dissociated and $i_{\text{OA}} = 1$.

To include effects of organic acid dissociation, $i_{\text{OA}}$ is calculated with consideration of the acid dissociation degree $\alpha$ (from eq. 26 and 30) as

$$i_{\text{OA}} = 1 + \alpha(n_{\text{ions}} - 1), \tag{34}$$

where $n_{\text{ions}} = 2$ for the monoprotic acid and $n_{\text{ions}} = 3$ for the diprotic acid, is the number of ions formed from one formula unit of the organic acid. The total available molar amount of solute ($n_s$, eq. 33) is thus modified by organic acid dissociation according to $i_{\text{OA}}$ from eq. 34 and reflected in the Raoult term $B$ (eq. 4) which changes the critical supersaturation $S_i$ for each aerosol size bin.

The van't Hoff factor is calculated for each $pK_a$ using eq. 34. The dissociation degrees and van't Hoff factors for the no organic dissociation (no diss), organic acid dissociation according to bulk acidity ($pK_a^{\text{bulk}}$) and surface modulated suppressed organic acid dissociation ($pK_a^{\text{bulk}}+1$ and $pK_a^{\text{bulk}}+2$), for all the OA mass fractions considered here are given in the Supplement (Table S1).

## 3   Results and discussions

We present the results of HAMBOX simulations for Sulfur chemistry, cloud microphysics, and aerosol-cloud-climate effects, considering organic acid bulk acidity ($pK_a^{\mathrm{bulk}}$), surface modulated suppressed organic acid dissociation ($pK_a^{\mathrm{bulk}}+1$ and 2), and no organic acid dissociation (no diss). Simulations were carried out with the entire OA fraction (Table 2) as either Malonic (OA = Malonic acid) or Decanoic (OA = Decanoic acid) acid. Results of Sulfur chemistry calculations are presented in terms of total Hydrogen ion concentration $[\mathrm{H}^+]_{\mathrm{tot}}$ and secondary Sulfate concentration $[\mathrm{SO}_4^{2-}]''$ in the aerosol population. The consecutive effect on cloud activating properties is then presented in terms of change in cloud droplet number concentration $\Delta$CDNC and cloud radiative effect RE predicted for bulk and surface modulated suppressed dissociation of the organic acids, compared to no organic dissociation.

### 3.1   Aqueous aerosol Hydrogen ion concentration

Figure 1 shows the total Hydrogen ion concentration $[\mathrm{H}^+]_{\mathrm{tot}}$ calculated with HAMBOX (eq. 20) in the aqueous aerosol population with sizes between $D_{\mathrm{wet}} = 0.317 - 40$ μm after 1 hour of simulation time, as a function of varying $pK_a$, corresponding to representations of bulk ($pK_a^{\mathrm{bulk}}$) and surface modulated ($pK_a^{\mathrm{bulk}}+1$ and 2) organic acid dissociation, considering OA = Malonic acid (panels a, b), and OA = Decanoic acid (panels c, d), for varying initial mass fraction of organic aerosol ($\chi_{\mathrm{OA}} = \{0.2, 0.4, 0.6, 0.8, 1\}$ in blue, purple, pink, orange, and yellow, respectively). The Hydrogen ion concentration with no organic dissociation (no diss, eq. 19) is also shown as a black line. As expected, $[\mathrm{H}^+]_{\mathrm{tot}}$ does not change with $pK_a$ when organic acid dissociation is not accounted for, whereas a significant increase is observed for both Malonic and Decanoic acids when organic acid dissociation is considered. The total Hydrogen ion concentration is highest when organic acid dissociation is considered according to $pK_a^{\mathrm{bulk}}$ and decreases for all $\chi_{\mathrm{OA}}$ as acid dissociation is increasingly suppressed according to the surface modulated apparent $pK_a^{\mathrm{bulk}}+1$ and $pK_a^{\mathrm{bulk}}+2$.

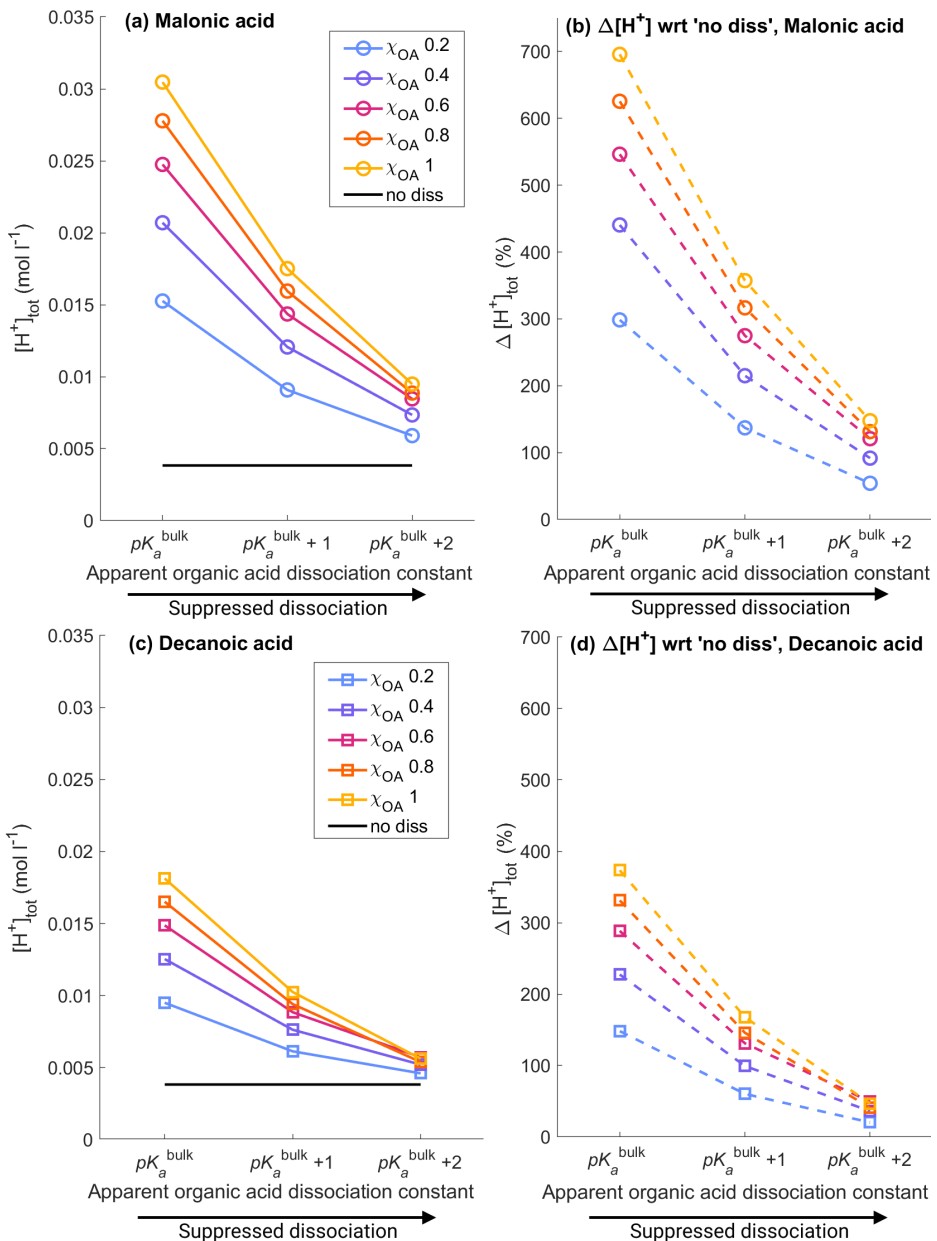

**Figure 1.** Aqueous aerosol total Hydrogen ion concentration, $[H^+]_{tot}$, calculated from eq. 20 with the Sulfur chemistry module of HAMBOX in the aqueous aerosol population with sizes between $D_{wet} = 0.317 - 40\,\mu m$, after 1 hour of simulation time, assuming five different initial organic mass fractions $\chi_{OA} = \{0.2, 0.4, 0.6, 0.8, 1\}$, denoted by blue, purple, pink, orange, and yellow, respectively (see also Table 2), with varying $pK_a$ corresponding to representations of bulk ($pK_a^{bulk}$) and surface modulated ($pK_a^{bulk} + 1$ and 2) organic acid dissociation. (a, b) All OA is assumed to be Malonic acid, and (c, d) all OA is assumed to be Decanoic acid. Simulations without accounting for organic acid dissociation are represented by 'no diss' and panels b and d show the relative change in total Hydrogen ion concentration $\Delta[H^+]_{tot}$ calculated from eq. 35 for each of the acid dissociation conditions with respect to 'no diss'.

The relative change in total Hydrogen ion concentration with respect to the 'no diss' condition

$$\Delta\left[\mathrm{H}^+\right]_{\mathrm{tot}} = \frac{[\mathrm{H}^+]_{\mathrm{tot}} - [\mathrm{H}^+]_0}{[\mathrm{H}^+]_0} \times 100, \tag{35}$$

is shown for Malonic and Decanoic acids in panels (b) and (d), respectively, of fig. 1. For Malonic acid with $pK_a^{\mathrm{bulk}}$, $\Delta[\mathrm{H}^+]_{\mathrm{tot}} = 298.5\%$ at the lowest OA mass fraction $\chi_{\mathrm{OA}} = 0.2$ and 696% for the highest OA mass fraction $\chi_{\mathrm{OA}} = 1$. Under the surface modulated suppressed organic acid dissociation condition $pK_a^{\mathrm{bulk}} + 1$, $\Delta[\mathrm{H}^+]_{\mathrm{tot}}$ decreases to 137% and 357% for $\chi_{\mathrm{OA}} = 0.2$ and $\chi_{\mathrm{OA}} = 1$, respectively. On further suppression of organic acid dissociation according to $pK_a^{\mathrm{bulk}} + 2$, $\Delta[\mathrm{H}^+]_{\mathrm{tot}}$ further decrease to 54% and 148%, respectively, at these OA mass fractions. For Decanoic acid organic aerosol, the total Hydrogen ion concentration at $pK_a^{\mathrm{bulk}}$ increases by 148% with respect to 'no diss' for $\chi_{\mathrm{OA}} = 0.2$ and by 374% for $\chi_{\mathrm{OA}} = 1$. At $pK_a^{\mathrm{bulk}} + 1$, $\Delta[\mathrm{H}^+]_{\mathrm{tot}}$ decreases to 61% and 168%, respectively, and at $pK_a^{\mathrm{bulk}} + 2$ further to 20% and 47% for $\chi_{\mathrm{OA}} = 0.2$ and 1, respectively. Therefore, even considering the stronger surface modulated suppression of organic acid dissociation with $pK_a^{\mathrm{bulk}} + 2$, the total Hydrogen ion concentration in the aqueous aerosol is still $20 - 47\%$ higher than 'no diss' for Decanoic acid, and $54 - 148\%$ higher for Malonic acid, depending on the initial organic aerosol mass fraction.

The results shown in fig. 1 and in the following are obtained for a constant ionic strength of $I = 0.5 \ \mathrm{mol\,kg}^{-1}$. Ionic strength is a bulk solution phenomenon and not expected to affect surface adsorbed organic acids, which can be considered as a (partially) liquid-liquid separated phase (Prisle et al., 2010a), to the same degree as in the bulk solution. Therefore, the amount of Hydrogen ions dissociated by the organic acid ($[\mathrm{H}^+]_{\mathrm{HA}}$, eq. 20) is expected to depend on $I$ mainly for the bulk solution condition and potentially to some extent for the surface modulated conditions. The total Hydrogen ion concentration ($[\mathrm{H}^+]_{\mathrm{tot}}$, eq. 20) in the aerosol population is shown in fig. S1 of the Supplement for OA = Malonic acid, considering $\chi_{\mathrm{OA}} = 0.4$ and 0.6, and for varying ionic strengths $I = \{0.5, 1, 3, 5\} \ \mathrm{mol\,kg}^{-1}$.

The total Hydrogen ion concentration decreases with increasing ionic strength, as expected. For $\chi_{\mathrm{OA}} = 0.4$, $[\mathrm{H}^+]_{\mathrm{tot}}$ is approximately 270% greater for $pK_a^{\mathrm{bulk}}$ than without consideration of dissociation (no diss) at $I = 5 \ \mathrm{mol\,kg}^{-1}$. For the surface modulated dissociation condition at the same ionic strength and organic aerosol mass fraction, $[\mathrm{H}^+]_{\mathrm{tot}}$ is approximately 120% greater for $pK_a^{\mathrm{bulk}} + 1$ compared to 'no diss'. Even for the more strongly suppressed dissociation corresponding to $pK_a^{\mathrm{bulk}} + 2$, $[\mathrm{H}^+]_{\mathrm{tot}}$ is approximately 45% higher than without dissociation. Therefore, for $I = \{0.5, 1, 3, 5\} \ \mathrm{mol\,kg}^{-1}$, the total Hydrogen ion concentration in the aqueous aerosol has significant contribution from organic acid dissociation. Similar analysis for varying ionic strength considering OA = Decanoic acid was not immediately possible, due to lack of data on the variation of $pK_a$ with $I$ for aqueous Decanoic acid solutions. However, measurements of $pK_a$ for Acetic acid in aqueous solutions with varying ionic strength were reported by Cohn et al. (1928). The total Hydrogen ion concentration for varying ionic strengths in the aqueous aerosol is shown for OA = Acetic acid in fig. S2 of the Supplement. For OA = Acetic acid, organic acid dissociation considering $I = \{0.02, 0.2, 1, 4\} \ \mathrm{mol\,kg}^{-1}$ results in approximately $270 - 380\%$ higher $[\mathrm{H}^+]_{\mathrm{tot}}$ than without acid dissociation. Both Acetic acid ($pK_a = 4.76$, Goldberg et al. (2002)) and Decanoic acid ($pK_a = 4.9$, Martell and Smith (1974)) are straight chain monocarboxylic acids with comparable bulk acidity and their aqueous dissociation properties are expected to be similar. Therefore, variation in $I$ is expected to result in similar $[\mathrm{H}^+]_{\mathrm{tot}}$ in the aqueous aerosol for OA = Decanoic acid as for OA = Acetic acid.

## 3.2 Aqueous aerosol Sulfate concentration

Figure 2 shows the aqueous phase secondary Sulfate concentrations $[SO_4^{2-}]''$ from oxidation of $SO_2$ by $H_2O_2$ and $O_3$ (Section 2.3) in the aqueous aerosol population with sizes between $D_{\text{wet}} = 0.317 - 40$ µm, after 1 hour of simulation time, for varying acid $pK_a$ corresponding to representations of bulk ($pK_a^{\text{bulk}}$) and surface modulated ($pK_a^{\text{bulk}} + 1$ and $pK_a^{\text{bulk}} + 2$) organic dissociation, assuming five different initial organic mass fractions $\chi_{\text{OA}} = \{0.2, 0.4, 0.6, 0.8, 1\}$, denoted by blue, purple, pink, orange, and yellow, respectively, and where OA = Malonic acid (panels a, c) and Decanoic acid (panels b, d). The secondary Sulfate concentration obtained from simulations without consideration of organic acid dissociation ('no diss', black line) is shown for reference. The relative changes in the Sulfate concentration compared to 'no diss' ($\Delta[SO_4^{2-}]''$) for both oxidation pathways are given in fig. S3 in the Supplement.

The Sulfate concentration from $H_2O_2$ oxidation (panels a, b) increases drastically for both organic acids when organic acid dissociation is accounted for. From eq. 15, it may seem that Sulfate concentration should decrease with increasing $[H^+]_{\text{tot}}$, but the reverse is observed in fig. 2. This is a property of the general acid catalysed mechanism, where the $pK_a$-dependent rate constant $k_{\text{HA}}$ offsets the decrease in $[SO_4^{2-}]''$ caused by increased $[H^+]_{\text{tot}}$ concentration. These results are in line with Liu et al. (2020), who suggested the general acid catalysed $H_2O_2$ oxidation to explain 'missing' Sulfate during severe haze episodes. The oxidation of $SO_2$ by $O_3$ (panels c, d) follows a straightforward dependence on $[H^+]_{\text{tot}}$ (eq. 16), where increased Hydrogen ion concentration results in decreased $[SO_4^{2-}]''$, compared to the 'no diss' condition.

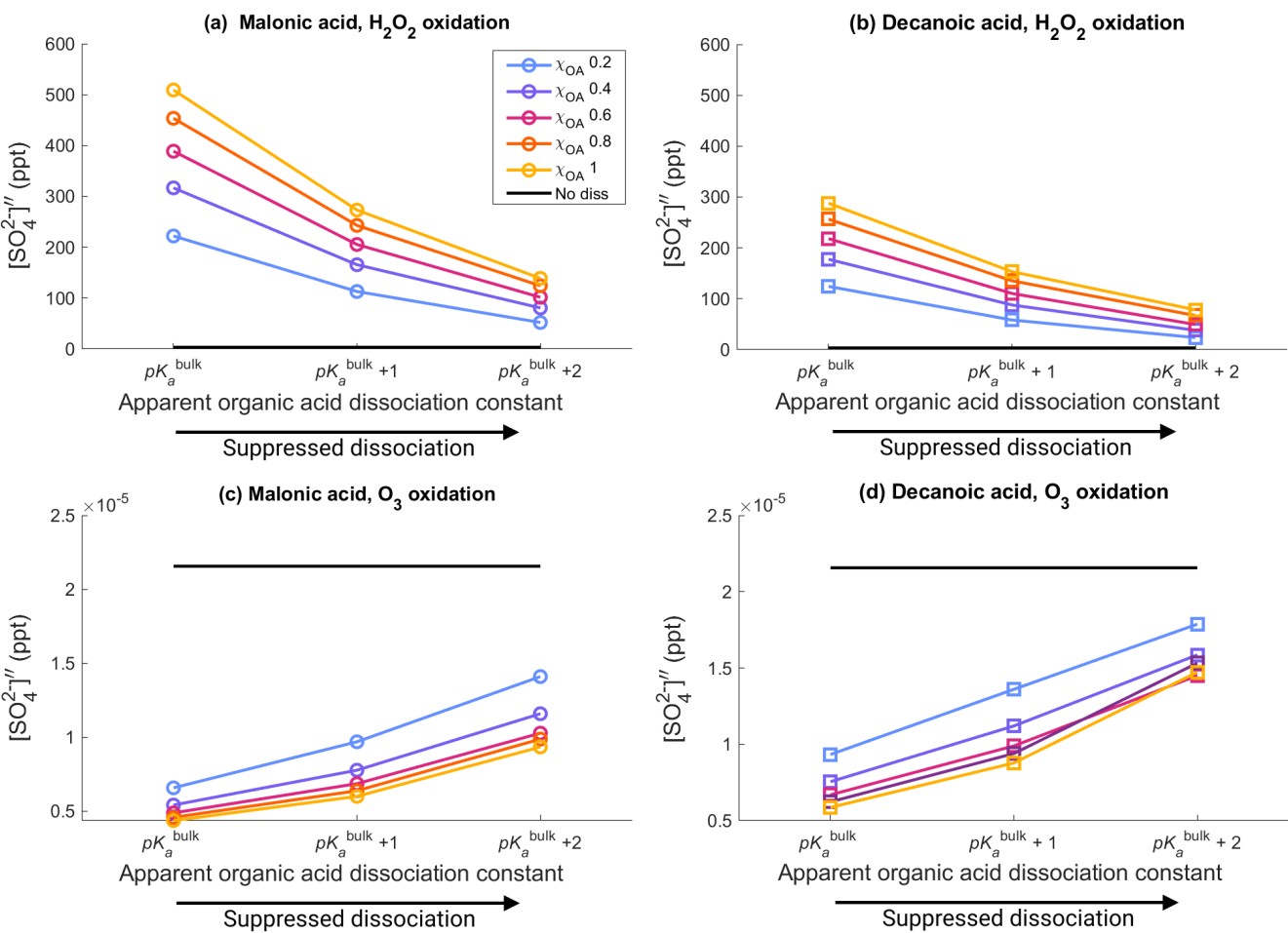

**Figure 2.** Aqueous aerosol secondary Sulfate concentrations $[SO_4^{2-}]''$ from oxidation of $SO_2$ by $H_2O_2$ (panels a, b, eqs. 15 and 13) and $O_3$ (panels c, d, eq. 16), for Malonic acid (panels a, c) and Decanoic acid (panels b, d), with varying $pK_a$ corresponding to representations of bulk ($pK_a^{bulk}$) and surface modulated ($pK_a^{bulk} + 1$ and 2) organic acid dissociation, in the aqueous aerosol population with sizes between $D_{wet} = 0.317 - 40$ μm, calculated in the Sulfur chemistry module of HAMBOX with five different initial organic mass fractions $\chi_{OA} = \{0.2, 0.4, 0.6, 0.8, 1\}$, denoted by blue, purple, pink, orange, and yellow, respectively (Table 2), after 1 hour of simulation time. Simulations without accounting for organic acid dissociation are represented by 'no diss' (black curves).

For Malonic acid, $H_2O_2$ oxidation shows an increase in aqueous phase secondary Sulfate concentration compared to 'no diss' (fig. 2 panel a), with $\Delta[SO_4^{2-}]''$ ranging from $6434 - 14876\%$ at $pK_a^{bulk}$ with increasing $\chi_{OA}$ (Supplement fig. S3 panel a). With surface modulated suppressed organic acid dissociation, $[SO_4^{2-}]''$ decreases compared to $pK_a^{bulk}$. The lowest $\Delta[SO_4^{2-}]''$ is predicted for $pK_a^{bulk} + 2$ at $\chi_{OA} = 0.2$. But even here, $\Delta[SO_4^{2-}]'' = 1432\%$, which is a strong increase compared to 'no diss'. Similar trends are found for the $H_2O_2$ oxidation with Decanoic acid (fig. 2 panel b), where the highest $[SO_4^{2-}]''$ is obtained for $pK_a^{bulk}$ with $\Delta[SO_4^{2-}]''$ ranging from $3557 - 8367\%$ with increasing $\chi_{OA}$ (Supplement fig. S3 panel b). The lowest $[SO_4^{2-}]''$ predicted for OA = Decanoic acid is seen for the stronger surface modulated suppression of organic acid dissociation

at $pK_a^{\text{bulk}} + 2$ and $\chi_{\text{OA}} = 0.2$, as expected, with $\Delta[\text{SO}_4^{2-}]'' = 598\%$. The aqueous phase secondary Sulfate concentration from $\text{O}_3$ oxidation of $\text{SO}_2$ (fig. 2 panel c), decreases by $70 - 80\%$ compared to 'no diss' for OA = Malonic acid at $pK_a^{\text{bulk}}$, with

increasing $\chi_{\text{OA}}$ (Supplement fig. S3 panel c). The decrease is smaller for surface modulated suppressed acid dissociation, as expected, with $\Delta[\text{SO}_4^{2-}]'' = 35 - 55\%$ for the more strongly suppressed dissociation at $pK_a^{\text{bulk}} + 2$. For OA = Decanoic acid (fig. 2 panel d), a similar trend is seen for $[\text{SO}_4^{2-}]''$ from $\text{O}_3$ oxidation, where decrease in aqueous phase secondary Sulfate concentration is in the range of $\Delta[\text{SO}_4^{2-}]'' = 20 - 75\%$ for bulk and surface modulated suppressed dissociation (Supplement fig. S3 panel d). Therefore, $\text{H}_2\text{O}_2$ oxidation of $\text{SO}_2$ results in a far greater increase in $[\text{SO}_4^{2-}]''$ in the aerosol population than

the decrease in $[\text{SO}_4^{2-}]''$ from the $\text{O}_3$ oxidation of $\text{SO}_2$, compared to 'no diss'.

These results show how the increase in $[\text{H}^+]_{\text{tot}}$ from organic acid dissociation in terms of $[\text{H}^+]_{\text{HA}}$ results in significant increases in predicted $[\text{SO}_4^{2-}]''$ in the aqueous aerosol, compared to when organic acid dissociation is not accounted for. As expected, the effect is smaller when organic dissociation is suppressed according to surface modulated $pK_a$, but even for the stronger suppression considered here, the effect is $1432 - 4000\%$ and $598 - 2500\%$ with increasing $\chi_{\text{OA}}$, for Malonic and

Decanoic acid organic aerosol, respectively. We see that the effect of acid dissociation is larger for $\text{H}_2\text{O}_2$ oxidation, suggesting that this pathway is more sensitive to inclusion of organic aerosol acidity and dissociation effects.

### 3.3 Activation of droplets

Figure 3 shows the critical supersaturation $S_i$ (eq. 3) predicted with HAMBOX for initial organic mass fractions $\chi_{\text{OA}} = \{0.2, 0.4, 0.6, 0.8, 1\}$, obtained for three dry particle sizes, $d_p = 135$ nm (darkest shade), $290$ nm (lighter shade) and $456$ nm

(lightest shade), with $pK_a$ represented by bulk ($pK_a^{\text{bulk}}$) and surface modulated ($pK_a^{\text{bulk}} + 1$ and $pK_a^{\text{bulk}} + 2$) organic acid dissociation, considering OA = Malonic acid (panel a), and OA = Decanoic acid (panel b). Results from simulations with the 'no diss' condition (black bars) are also shown in both panels for each dry particle size. The dry particle sizes chosen here fall in the Accumulation sub-range of size bins. We choose these dry particle sizes to investigate the critical supersaturation as particles in the Accumulation size range have been shown to be the most effective in CDNC production (Patel and Jiang,

2021).

The redistribution of aerosol sizes caused by the increased Sulfate concentrations affects the calculation of the droplet size $D_{\text{wet}}$ (eq. 3). Therefore, $S_i$ is affected by bulk ($pK_a^{\text{bulk}}$) and surface modulated ($pK_a^{\text{bulk}} + 1$ and $pK_a^{\text{bulk}} + 2$) acid dissociation of the OA. We see that for both organic acids considered, the increased secondary Sulfate concentrations in the aqueous aerosols, compared to 'no diss', is sufficient to significantly decrease $S_i$ for all $\chi_{\text{OA}}$ at each of the three dry particle sizes considered.

As expected, the decrease in $S_i$ is smaller when surface modulated suppressed acid dissociation is considered, compared to simulations considering bulk acidity of OA. For OA = Decanoic acid (panel b), the difference between $S_i$ calculated for bulk and surface modulated $pK_a$ is larger than for OA = Malonic acid (panel a), especially for the larger dry particle sizes (lightest shade) and higher organic mass fractions, $\chi_{\text{OA}} = 0.6 - 1$. This suggests that $S_i$ from consideration of Decanoic acid dissociation is more susceptible to changes in the surface modulated $pK_a$ than for Malonic acid, especially at higher $d_p$ and $\chi_{\text{OA}}$. While

Decanoic acid is the more surface active of the two organic acids considered, because our simple empirical representation does not explicitly account for surface adsorption, this effect is here caused by the differences in bulk and apparent surface

modulated $pK_a$ with respect to the aerosol pH. As the organic mass fraction increases, the difference in $S_i$ between all $pK_a$ conditions and 'no diss' increases for both organic acids.

The calculated $S_i$ in each size bin also includes any changes in aerosol water activity due to increased organic van't Hoff factor from organic acid dissociation (eq. 34, Section 2.4.4). We see that for both organic acids, the water activity is sufficiently reduced, even for the surface modulated suppressed acid dissociation, to significantly decrease $S_i$, compared to 'no diss'. It is well established that aerosol critical supersaturation typically increases with increasing OA mass fraction, due to the higher hygroscopicity of organic aerosol components compared to other soluble species, such as inorganic salts (Svenningsson et al., 2006). However, fig. 3 shows that organic acid dissociation can partially counter this increase in $S_i$. This effect is smaller if surface modulated suppression of organic acid dissociation is considered, which is expected to be more relevant for smaller particles and droplets, due to their high surface area to bulk volume ratio, and for aerosol populations with higher fractions of surface active OA (Prisle, 2021).

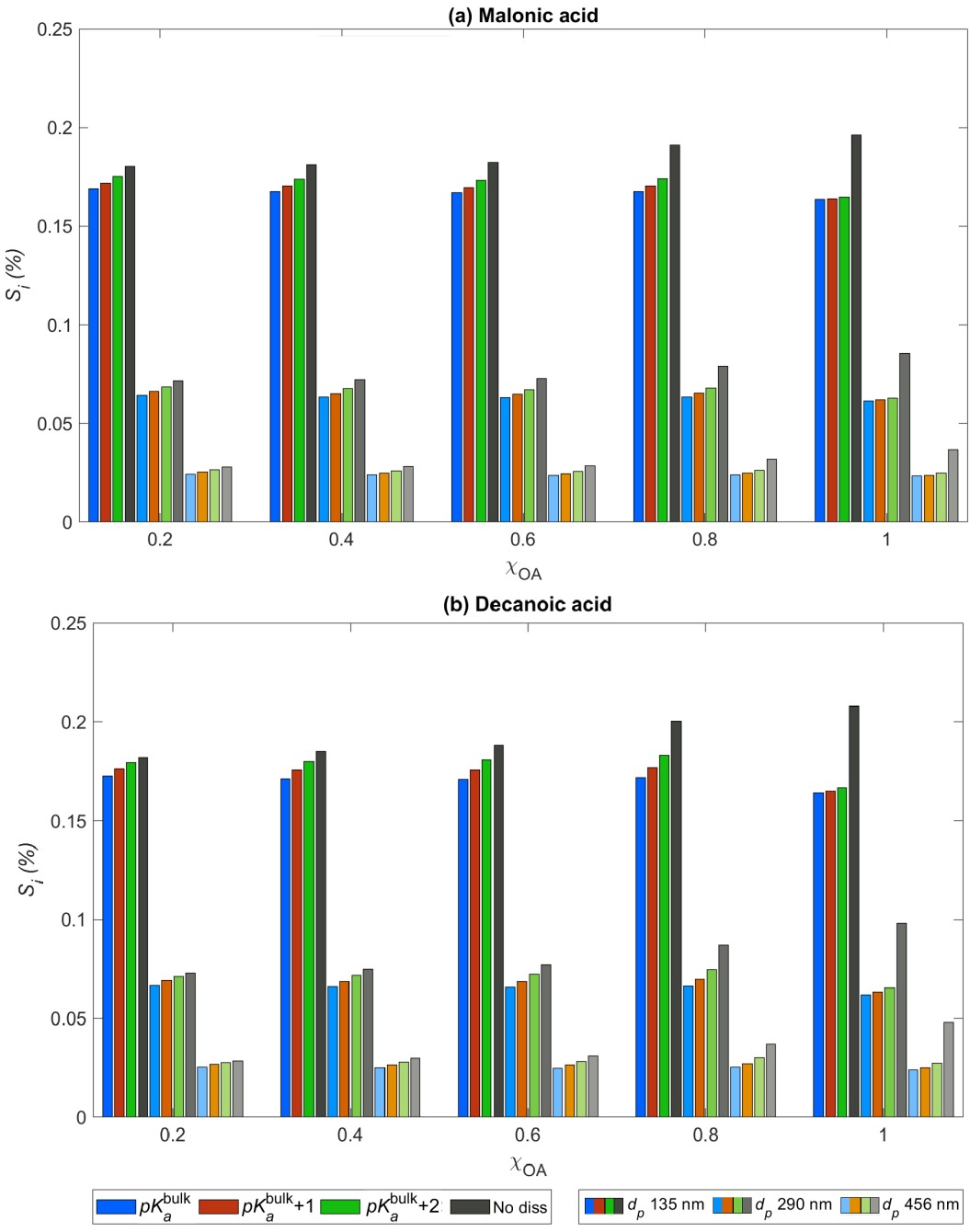

**Figure 3.** Critical supersaturation $S_i$ as a function of the initial organic aerosol mass fraction $\chi_{OA}$ for (a) Malonic acid and (b) Decanoic acid, for three initial dry particle sizes, $d_p$ = 135 nm (darkest shade), 290 nm (lighter shade), and 456 nm (lightest shade), and $pK_a$ representing bulk ($pK_a^{bulk}$, in blue) and surface ($pK_a^{bulk}+1$, in orange, and $pK_a^{bulk}+2$, in green) organic acid dissociation. Simulations without accounting for organic acid dissociation are represented by 'no diss' (black and grey bars).

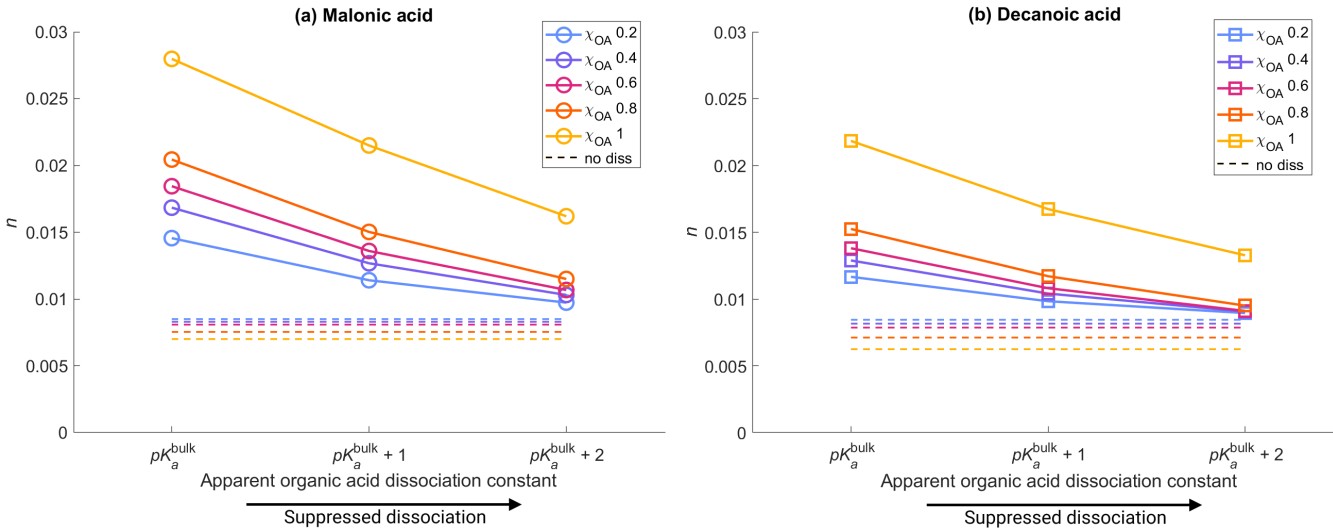

**Figure 4.** The average activated fraction, $n$, of the aerosol population with droplet sizes between $D_{\mathrm{wet}} = 0.317 - 40$ μm, after 1 hour of simulation time, for five different initial organic mass fractions $\chi_{\mathrm{OA}} = \{0.2, 0.4, 0.6, 0.8, 1\}$, denoted by blue, purple, pink, orange, and yellow, respectively (see also Table 2), calculated for varying $pK_a$ corresponding to representations of bulk ($pK_a^{\mathrm{bulk}}$) and surface modulated ($pK_a^{\mathrm{bulk}} + 1$ and 2) organic acid dissociation, and considering (a) OA = Malonic acid, and (b) OA = Decanoic acid. The average activated fraction in the 'no diss' condition is shown as dashed lines in corresponding colors for each $\chi_{\mathrm{OA}}$.

Figure 4 shows the activated fraction $n$ (eq. 8) of the aerosol population, with droplet sizes between $D_{\mathrm{wet}} = 0.317 - 40$ μm, averaged for all aerosol size bins, calculated for initial organic mass fractions $\chi_{\mathrm{OA}} = \{0.2, 0.4, 0.6, 0.8, 1\}$, denoted by
blue, purple, pink, orange, and yellow, respectively (Table 2), with $pK_a$ representing bulk ($pK_a^{\mathrm{bulk}}$) and surface ($pK_a^{\mathrm{bulk}} + 1$ and $pK_a^{\mathrm{bulk}} + 2$) organic acid dissociation, considering OA = Malonic acid (panel a) and OA = Decanoic acid (panel b). The average activated fractions for simulations not considering organic acid dissociation ('no diss') are shown as dashed lines in corresponding colors for each $\chi_{\mathrm{OA}}$, approximately $0.006 - 0.008$ for both organic acids. The inclusion of organic acid dissociation effects results in higher activated fractions than 'no diss' for both organic acids, with Malonic acid dissociation resulting in a greater $n$ than Decanoic acid, for the same $\chi_{\mathrm{OA}}$ and $pK_a$. This is expected, as Malonic acid is a stronger acid with lower $pK_a^{\mathrm{bulk}}$ and $[\mathrm{H}^+]_{\mathrm{tot}}$ from Malonic acid dissociation under the same conditions is higher than for Decanoic acid. The maximum activated fraction is observed for $pK_a^{\mathrm{bulk}}$ with $n = 0.014 - 0.027$ for OA = Malonic acid and $n = 0.012 - 0.021$ for OA = Decanoic acid. For surface modulated suppressed organic acid dissociation according to $pK_a^{\mathrm{bulk}} + 1$, the activated fraction decreases to $0.011 - 0.021$ and $0.010 - 0.016$ for OA = Malonic and Decanoic acids, respectively. As expected, the stronger surface modulated acid dissociation suppression according to $pK_a^{\mathrm{bulk}} + 2$ further decreases the activated fraction, but $n$ is still higher than for 'no diss' ($0.009 - 0.016$ and $0.009 - 0.013$ for OA = Malonic and Decanoic acids, respectively). For both organic acids, the activated fraction also increases with increasing $\chi_{\mathrm{OA}}$, which is expected as the $[\mathrm{H}^+]_{\mathrm{tot}}$ increases with increasing $\chi_{\mathrm{OA}}$ (fig 1). For both organic acids, $[\mathrm{H}^+]_{\mathrm{tot}}$ is sufficiently high to lead to a decrease in $S_i$ that translates into

an increased activated fraction for both bulk and surface modulated suppressed organic acid dissociation conditions, with a
smaller effect for the suppressed acid dissociation, as expected.

### 3.4 Cloud droplet number concentration

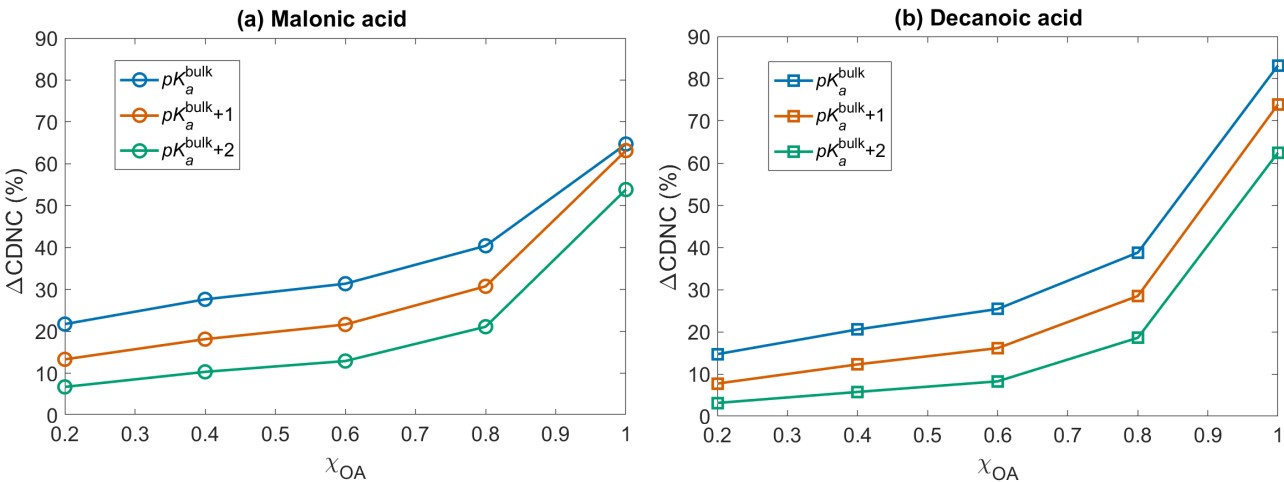

**Figure 5.** The change in cloud droplet number concentration $\Delta$CDNC (eq. 10) with respect to 'no diss' for (a) OA = Malonic acid and
(b) OA = Decanoic acid, as a function of initial organic mass fractions $\chi_{OA} = \{0.2, 0.4, 0.6, 0.8, 1\}$ assuming organic acid dissociation
according to bulk ($pK_a^{bulk}$, blue) and surface modulated properties ($pK_a^{bulk} + 1$, orange, and $pK_a^{bulk} + 2$, green).

Figure 5 shows the change in cloud droplet number concentration $\Delta$CDNC (eq. 10) with respect to 'no diss', as a function of
initial organic mass fractions $\chi_{OA} = \{0.2, 0.4, 0.6, 0.8, 1\}$, considering OA = Malonic acid (panel a), and OA = Decanoic acid
(panel b), for varying $pK_a$ corresponding to representations of bulk ($pK_a^{bulk}$, in blue), and surface modulated ($pK_a^{bulk} + 1$,
in orange, and $pK_a^{bulk} + 2$, in green) organic acid dissociation. A significant enhancement in CDNC is seen for both Malonic
and Decanoic acid, compared to when no organic acid dissociation is considered (no diss). Similar trends are seen for both
acids, where $pK_a^{bulk}$ shows the highest CDNC enhancement compared to 'no diss'. This is expected based on the calculated
$[H^+]_{tot}$ (fig. 1) and $[SO_4^{2-}]''$ (fig. 2) from the bulk organic acidity and surface modulated suppressed organic acid dissociation
and consequent critical supersaturation ($S_i$, fig. 3) from both organic acids. For $pK_a^{bulk}$ and OA = Decanoic acid, $\Delta$CDNC
ranges from 14.7% to 83.1% with increasing $\chi_{OA}$. For OA = Malonic acid, the $\Delta$CDNC is smaller, ranging from 21.7% to
64.7% for corresponding $\chi_{OA}$ considering bulk acidity $pK_a^{bulk}$. Under surface modulated suppressed organic acid dissociation
$pK_a^{bulk} + 1$, the CDNC enhancement is less than that obtained from $pK_a^{bulk}$, ranging from 7.7% to 73.9% for Decanoic acid
and 13.3% to 63.1% for Malonic acid. For the stronger surface modulated acid dissociation suppression $pK_a^{bulk} + 2$, $\Delta$CDNC
is 3.1% to 62.5% for OA = Decanoic acid and 6.7% to 53.9% for OA = Malonic acid, with increasing $\chi_{OA}$.

The CDNC enhancement upon including OA acid dissociation is caused by the change in aerosol size distribution due to
the increased Sulfate concentrations, which shifts the distribution towards larger particles, which are more effective in CDNC

production (Hudson and Da, 1996; McFiggans et al., 2006). Since size plays a significant role in cloud droplet nucleating ability of aerosol particles (Dusek et al., 2006), the effect of organic acid dissociation on cloud response will be different depending on whether bulk or surface modulated properties are used to describe the organic aerosol, and from fig. 5 we see that this difference is significant for both Malonic and Decanoic acid under the simulation conditions. The aerosol size distribution after one hour of simulation time with and without activating the Sulfur chemistry module, is given in the Supplement (figs. S4 and S5) for both organic acids, assuming initial organic mass fraction, $\chi_{OA} = 0.8$. The aerosol size distribution is shown for varying $pK_a$ corresponding to representations of bulk ($pK_a^{bulk}$) and surface modulated ($pK_a^{bulk} + 1$ and 2) organic acid dissociation, together with the no acid dissociation condition. Without organic acid dissociation, the size distribution is almost the same after one hour for simulations with and without activating the Sulfur chemistry module, for both organic acids. For bulk acidity $pK_a^{bulk}$, the aerosol size distribution at one hour is significantly different from the 'no diss' size distribution. The change is smaller for the suppressed organic acid dissociation conditions $pK_a^{bulk} + 1$ and $pK_a^{bulk} + 2$. However, even for the stronger suppressed dissociation the change in aerosol size distribution is sufficient to yield $\Delta$CDNC by 6.7–53.8 % and 3.1–62.4 % for OA = Malonic acid and Decanoic acid, respectively, compared to 'no diss'.

### 3.5 Short-wave radiative effect

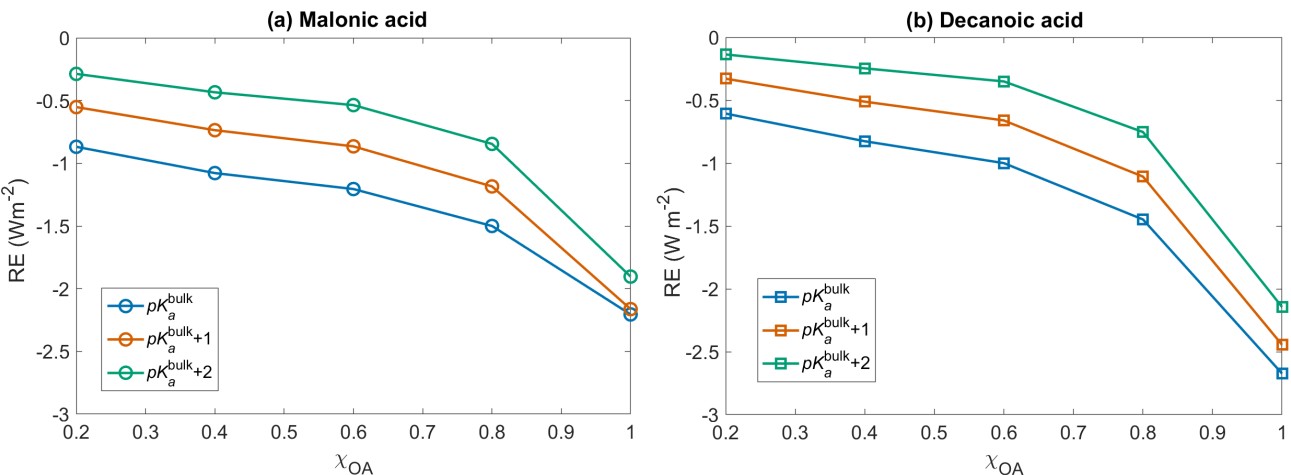

**Figure 6.** Short wave radiative effect (RE, eq. 12) with respect to 'no diss' for (a) Malonic acid and (b) Decanoic acid, for varying initial organic mass fractions ($\chi_{OA} = 0.2 - 1$) assuming organic dissociation according to bulk acidity ($pK_a^{bulk}$, blue) and surface modulated organic acid dissociation ($pK_a^{bulk} + 1$, orange, and $pK_a^{bulk} + 2$, green).

Figure 6 shows the short-wave radiative effect RE (eq. 12) with respect to 'no diss', as a function of initial organic mass fractions $\chi_{OA} = 0.2 - 1$, considering OA = Malonic acid (panel a), and OA = Decanoic acid (panel b), for varying $pK_a$ corresponding to representations of bulk ($pK_a^{bulk}$, in blue), and surface modulated ($pK_a^{bulk} + 1$, in orange, and $pK_a^{bulk} + 2$, in green) organic acid dissociation. The inclusion of organic acid dissociation leads to a cooling effect for both organic acids, compared to 'no

diss'. Considering organic acid dissociation with bulk properties ($pK_a^{\text{bulk}}$), a larger cooling effect is observed for both acids, ranging from $-0.6 \text{ W m}^{-2}$ to $-2.7 \text{ W m}^{-2}$ for Decanoic acid and $-0.86 \text{ W m}^{-2}$ to $-2.2 \text{ W m}^{-2}$ for Malonic acid, as $\chi_{\text{OA}}$ increases from 0.2 to 1. The effect is smaller when surface modulated suppression of acid dissociation is considered, but still significant compared to 'no diss'. For OA = Malonic acid, the range of RE extends from $-0.5 \text{ W},\text{m}^{-2}$ to $-2.2 \text{ W},\text{m}^{-2}$ for $pK_a^{\text{bulk}}+1$, and from $-0.3 \text{ W},\text{m}^{-2}$ to $-1.6 \text{ W},\text{m}^{-2}$ for the more strongly suppressed organic acid dissociation $pK_a^{\text{bulk}}+2$. For OA = Decanoic acid, the range of RE varies from $-0.3 \text{ W},\text{m}^{-2}$ to $-2.4 \text{ W},\text{m}^{-2}$ for $pK_a^{\text{bulk}}+1$, and from $-0.1 \text{ W},\text{m}^{-2}$ to $-2.1 \text{ W},\text{m}^{-2}$ for $pK_a^{\text{bulk}}+2$.

The effects of OA acid dissociation on cloud droplet number concentrations and radiative effect without considering the changes in the aqueous aerosol Sulfur chemistry are shown in fig. S6 of the Supplement. Here, the effects of organic acid dissociation arise from modification of the van't Hoff factor $i_{\text{OA}}$ (eq. 34), reflected as changes in the Raoult term $B$ (eq. 4) and consequently $S_i$ (eq. 3). The $\Delta$CDNC and RE in fig. S6 are therefore independent of the increased [H$^+$] driven Sulfate concentrations in the aqueous aerosol and instead reflect the change in water activity due to the dissociation of the organic acid and consequent increase in the number of available moles of solute $n_s$ (eq. 33). Considering OA acid dissociation effects in the water activity exclusively, OA = Malonic acid shows $< 0.5\%$ $\Delta$CDNC with respect to 'no diss' for all representations of organic acid dissociation. For OA = Decanoic acid, $\Delta$CDNC with respect to 'no diss' is slightly higher ($\approx 1\%$), specially for bulk acidity $pK_a^{\text{bulk}}$ and higher $\chi_{\text{OA}}$. The resulting RE with respect to 'no diss' is within a range between 0 and $-0.01$ W/m$^{-2}$ for OA = Malonic acid and 0 to $-0.05$ W/m$^{-2}$ for OA = Decanoic acid.

## 3.6 Discussion

Our results show that acid dissociation of organic aerosols, exemplified with Malonic and Decanoic acid as common atmospheric moderately strong acids, can influence aqueous phase Sulfur chemistry to significantly impact the cloud short-wave radiative effect. Surface modulated suppressed acid dissociation of OA can further change the concentration of Hydrogen ions in aqueous aerosol from what is immediately expected from the bulk acidity and aerosol pH. Under these surface modulated conditions, the effects of organic acid dissociation on cloud properties are reduced, but still significant. Since many components in atmospheric OA are both acidic and surface active, this may be important to represent in large scale models. Furthermore, the impact of OA acid dissociation on cloud activating properties via aqueous phase aerosol Sulfur chemistry is significantly stronger than by changing the aerosol water activity and will be strongly underestimated if only effects on water activity are considered.

The increased Hydrogen ion concentration in aqueous aerosols as a result of OA acid dissociation leads to enhanced Sulfate mass from oxidation of SO$_2$ by H$_2$O$_2$. Increased Sulfate concentrations could potentially lead to enhanced formation of organosulfur compounds within the aerosols. Organosulfur compounds form in the atmosphere through heterogeneous reactions between volatile organic compounds and inorganic aerosol Sulfate and can comprise over 15% of the secondary organic aerosol mass (Brüggemann et al., 2020; Riva et al., 2019; Chen et al., 2021a; Hettiyadura et al., 2019). These organosulfur compounds could further increase OA mass and affect the resulting cloud droplet number concentrations. Organosulfates have been shown to exhibit acidic properties and primarily exist in the deprotonated form under atmospheric pH conditions

(Fankhauser et al., 2022). Organosulfates are also known to be surface active in aqueous solutions (Hansen et al., 2015; Prisle et al., 2010b, 2011; Lin et al., 2018; Malila and Prisle, 2018). Therefore, the mechanisms of both organic acid dissociation and its surface modultation studied here for atmospheric carboxylic acids could also apply to organosulfate aerosols, potentially affecting cloud droplet activation in a similar manner.

Bulk–surface partitioning in aqueous aerosols can be seen as a form of (potentially second order) liquid-liquid phase separation (Prisle, 2023). Phase separation of organic aerosols and its impact on cloud activating properties of aerosol particles have been widely studied (Reid et al., 2018; Freedman, 2017; You et al., 2014). The partitioning of surface active aerosol components occurs between the bulk and surface phases due to differences in composition and affinity for each phase. The suppression of organic acid dissociation considered here is exactly a consequence of the increased concentration of surface active organic acid in the surface phase. Liquid-liquid phase separation in the bulk phase would effectively create two separate solutions with different compositions and ensuing properties. The modulation of organic acid dissociation could be taken into account separately for these phases, based on their individual concentrations, following analogous schemes as described by Prisle et al. (2010a).

Our results suggest that organic acid dissociation should be considered for accurate predictions of OA chemistry and cloud microphysics in the atmosphere. The specific magnitude of predictions with the present box model implementation may not be immediately representative of analogous simulations with full 3D aerosol-chemistry-climate models, due to their greater complexity and numerous coupled processes. For example, the effect of potential surface modulated suppressed dissociation will further depend on aerosol surface activity and is expected to be especially relevant for smaller droplet sizes. However, as we have used a box model version of ECHAM-HAMMOZ, implementation of the OA acid dissociation mechanisms considered here will follow analogous strategies for the full model. The box model simulations contribute insights into the detailed mechanisms of OA acid dissociation and its impact on aerosol chemistry and cloud formation and our present results provide a first assessment of the potential significance for resulting aerosol-cloud-climate parameters under conditions similar to those examined here.

## 4 Conclusion

We investigated the effects of organic aerosol acid dissociation on total Hydrogen ion concentration in aqueous aerosols and the impact on resulting secondary Sulfate aerosol mass, cloud droplet number concentration, and aerosol short-wave radiative effect, using the aerosol–chemistry-climate box model ECHAM6.3–HAM2.3 (HAMBOX). Simulations were carried out considering the entire OA to comprise organic acid and we used Malonic and Decanoic acid as proxies for atmospheric OA with different aqueous acidity and surface activity. Dissociation of organic acids was considered in three scenarios: 1) the current standard of no dissociation, 2) following well known bulk solution acidity given by the reported acid constant $pK_a^{\text{bulk}}$, and 3) accounting for a surface-modulated suppression of dissociation as observed in recent laboratory experiments.

Our results show that organic acid dissociation increases Hydrogen ion concentrations in the aqueous aerosol phase, as expected. This leads to strongly increased secondary Sulfate aerosol mass, which in turn decreases the critical supersaturation

for cloud droplet activation and yields a higher activated aerosol fraction than if OA acid dissociation is not considered. The cloud response is observed as enhanced cloud droplet number concentration and a strong short-wave radiative effect of clouds. The effects of organic acid dissociation are greatest when considering the bulk acidity of OA, but still significant even when potential surface-modulated suppression of dissociation is also included.

As many atmospheric organic aerosol components are acidic (Pye et al., 2020), their dissociation can have significant impacts on cloud properties. This work highlights the importance of including such organic acid dissociation effects in large scale atmospheric models. We suspect that, combined with the high surface area to bulk volume ratio and bulk–surface partitioning in small droplets (Bzdek et al., 2020; Prisle, 2021), the effects of organic dissociation and potential size dependent surface modulated acid dissociation could be significant in explaining some knowledge gaps about organic aerosol formation and acidity in atmospheric aerosols. OA acid dissociation could be particularly relevant in explaining discrepancies of atmospheric models with observations for polluted environments (Lee et al., 2013), where organic mass fraction is usually high and the organic acid dissociation effects could become very significant. Many of these organic aerosol acids may also be surface active in aqueous solutions (Gérard et al., 2019b), such as haze and activating cloud droplets, and therefore corrections to account for surface modulated suppression of organic acid dissociation may also be necessary.

*Code and data availability.* The HAMBOX code is available from the HAMMOZ Redmine here. All simulation data and scripts underlying the figures are available here

.

*Author contributions.* GS did the model implementations and performed the calculations with contributions from MZ. GS and NLP analyzed the results and wrote the original and revised manuscripts and response to reviewers. NLP conceived, planned, supervised, and secured funding for the project. All authors approved the final text.

*Competing interests.* The authors declare that there is no conflict of interest.

*Acknowledgements.* The authors warmly thank Kunal Ghosh and Harri Kokkola for valuable support on HAMBOX. This project has received funding from the European Research Council (ERC) under the European Union's Horizon 2020 research and innovation program, project SURFACE (grant agreement no. 717022). The authors also gratefully acknowledge the financial contribution from the Academy of Finland, including grant nos. 308238, 314175, and 335649.

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
