# Peer review of "Impact of acidity and surface modulated acid dissociation on cloud response to organic aerosol"

_EGUsphere, 2023_

## Author Comment (AC1)

**Author response to reviewers' comments**

Gargi Sengupta[a], Minjie Zheng[a,b], and Nønne L. Prisle[a*]

[a]Center for Atmospheric Research, University of Oulu, P.O. Box 4500, 90014 Oulu, Finland
[b]Current affiliation: Institute for Atmospheric and Climate Science, ETH Zurich, Zürich, Switzerland

**Correspondence:** Nønne L. Prisle (Nonne.Prisle@oulu.fi)

We thank both reviewers for their careful revision of our manuscript and constructive comments. Below we provide our responses in a point-by-point manner. The reviewers' comments are reproduced in *italics*, our responses in blue and quotes from the revised manuscript in **red bold** font. In addition, we have made a few minor edits to the manuscript in places not directly indicated by the reviewers, to further clarify some of the points.

**1  Reviewer #1**

**1.1  General comments**

*Sengupta et al. investigated the impact of surface modulated organic acid dissociation (and related acidity change) in organic aerosol on aqueous-phase SO2 oxidation, cloud responses, and radiative effects using a box model. This is an interesting topic and the authors clearly showed that the impacts could be significant. However, how this result translates into a 3D aerosol-chemistry-climate model (not a box-model version) is unclear. I think that the authors should be clearer about the goal of this paper. Is it just to demonstrate that these impacts can be significant, or to improve aerosol-chemistry-climate models by providing information about this previously largely overlooked aspect? If it is closer to the latter, how to implement the results of this paper in an aerosol-chemistry-climate model is key information needed (e.g., how to set up similar calculations in aerosol-chemistry-climate models or what are the recommended values for relevant key parameters in aerosol-climate simulations). Otherwise, the authors should make it very clear that the results of this paper are not ensured to be transferable to real-world aerosol-chemistry-climate simulations. Even so, more could be done to show the implications of these box-model-based results for aerosol-chemistry-climate simulations.*

The goal of our work is to demonstrate the potential impact of organic acid dissociation for aerosol-cloud interactions. We show that organic aerosol dissociation and its possible surface modulation for surface active organic acids can significantly impact cloud formation and aerosol-cloud-climate parameters. By demonstrating the detailed mechanism in a box model, we pave the way for subsequent implementation in large-scale atmospheric models. We use a box model ECHAM6.3–HAM2.3 (HAMBOX) which is part of a full 3D aerosol-chemistry-climate model ECHAM–HAMMOZ, and therefore implementation will follow a similar simulation setup. This is the topic of ongoing work by some of the authors and will be further detailed in a separate manuscript.

We have clarified the purpose and focus of the work in the Introduction of the revised manuscript:

**"In this work, we use an aerosol–chemistry-climate box model to investigate the potential impact on aerosol forming aqueous phase Sulfur chemistry, cloud droplet activation, and aerosol-cloud-climate parameters of organic acid dissociation in aqueous aerosols and its additional surface modulation for surfactant acidic OA."**

We have also addressed the transferrability of results from the box model implementation to large-scale global climate models in the Results and discussions Section 3.6 in the revised manuscript:

**"Our results suggest that organic acid dissociation should be considered for accurate predictions of OA chemistry and cloud microphysics in the atmosphere. The specific magnitude of predictions with the present box model implementation may not be immediately representative of analogous simulations with full 3D aerosol-chemistry-climate models, due to their greater complexity and numerous coupled processes. For example, the effect of potential surface modulated suppressed dissociation will further depend on aerosol surface activity and is expected to be especially relevant for smaller droplet sizes. However, as we have used a box model version of ECHAM-HAMMOZ, implementation of the OA acid dissociation mechanisms considered here will follow analogous strategies for the full model. The box model simulations contribute insights into the detailed mechanisms of OA acid dissociation and its impact on aerosol chemistry and cloud formation and our present results provide a first assessment of the potential significance for resulting aerosol-cloud-climate parameters under conditions similar to those examined here."**

**1.2 Specific comments**

*Section 2.3: does inorganic ion surface propensity play a role in SO2 oxidation here? What about ionic strength, as Liu et al. (2020) showed to substantially affect this chemistry.*

Surface propensity of inorganic ions would in principle also change aqueous droplet bulk composition via similar mechanisms as seen for organic surface activity (Li et al., 2017; Peychev and Slavchov, 2023). However, the effect is expected to be much smaller for inorganic ions than for surface active organics, which can be strongly depleted from the bulk phase in submicron droplets (Prisle et al., 2010b) Furthermore, the synergy effects of organic and inorganic surface adsorption are currently not well known. Therefore, it is not straightforward to include these effects convincingly in our simulations. We emphasize that other effects of organic partitioning have not been explicitly considered in this implementation. We only consider here the effects on overall droplet acidity due to the modulation of dissociation for organic acids adsorbed in the surface, according to the previous observations in surface sensitive X-ray photoelectron spectroscopy experiments (Werner et al., 2018; Prisle, 2023). This is because our main aim of this work is to specifically investigate the effects of OA acid dissociation and its potential surface modulation on aerosol-cloud interactions.

60

The ionic strength of aqueous droplets does affect the sulfate production chemistry. For this work, we chose a constant ionic strength, $I = 0.5 \ \mathrm{mol \ kg^{-1}}$, because this is the condition for which the modified sulfur chemistry mechanism was defined for all organic acids by Liu et al. (2020). The rate constant $k_{\mathrm{HA}}$ (eq. 15, revised manuscript) depends on the ionic strength and its relationship with $pK_a$ of a given organic acid was given by Liu et al. (2020) considering an ionic strength of $0.5 \ \mathrm{mol \ kg^{-1}}$.

65 Since detailed variation of rate constants for these reactions were not available in literature for other ionic strengths, we assume $I = 0.5 \ \mathrm{mol \ kg^{-1}}$ in all our calculations. We have clarified this in Methods Section 2.3 of the revised manuscript by adding:

**"This approximation for $k_{\mathrm{HA}}$ in relation to the $pK_a$ of an organic acid was derived by Liu et al. (2020) for an ionic strength of $I = 0.5 \ \mathrm{mol \ kg^{-1}}$. Therefore, we assume this same ionic strength for aqueous droplets in all our calcula-**

70 **tions."**

Under varying ionic strength, the amount of hydrogen ions dissociated by the organic acid and the consequent sulfur chemistry will also vary. We have included a discussion on this in the Results and discussions Section 3.1 of the revised manuscript:

75 **"The results shown in fig. 1 and in the following are obtained for a constant ionic strength of $I = 0.5 \ \mathrm{mol \ kg^{-1}}$. Ionic strength is a bulk solution phenomenon and not expected to affect surface adsorbed organic acids, which can be considered as a (partially) liquid-liquid separated phase (Prisle et al., 2010a), to the same degree as in the bulk solution. Therefore, the amount of Hydrogen ions dissociated by the organic acid ($[\mathrm{H^+}]_{\mathrm{HA}}$, eq. 20) is expected to depend on $I$ mainly for the bulk solution condition and potentially to some extent for the surface modulated conditions. The total**

80 **Hydrogen ion concentration ($[\mathrm{H^+}]_{\mathrm{tot}}$, eq. 20) in the aerosol population is shown in fig. S1 of the Supplement for OA = Malonic acid, considering $\chi_{\mathrm{OA}} = 0.4$ and $0.6$, and for varying ionic strengths $I = \{0.5, 1, 3, 5\} \ \mathrm{mol \ kg^{-1}}$.**

**The total Hydrogen ion concentration decreases with increasing ionic strength, as expected. For $\chi_{\mathrm{OA}} = 0.4$, $[\mathrm{H^+}]_{\mathrm{tot}}$ is approximately $270\%$ greater for $pK_a^{\mathrm{bulk}}$ than without consideration of dissociation (no diss) at $I = 5 \ \mathrm{mol \ kg^{-1}}$. For the surface modulated dissociation condition at the same ionic strength and organic aerosol mass fraction, $[\mathrm{H^+}]_{\mathrm{tot}}$ is ap-**

85 **proximately $120\%$ greater for $pK_a^{\mathrm{bulk}} + 1$ compared to 'no diss'. Even for the more strongly suppressed dissociation corresponding to $pK_a^{\mathrm{bulk}} + 2$, $[\mathrm{H^+}]_{\mathrm{tot}}$ is approximately $45\%$ higher than without dissociation. Therefore, for $I = \{0.5, 1, 3, 5\}$ $\mathrm{mol \ kg^{-1}}$, the total Hydrogen ion concentration in the aqueous aerosol has significant contribution from organic acid dissociation. Similar analysis for varying ionic strength considering OA = Decanoic acid was not immediately possible, due to lack of data on the variation of $pK_a$ with $I$ for aqueous Decanoic acid solutions. However, measurements of**

90 **$pK_a$ for Acetic acid in aqueous solutions with varying ionic strength were reported by Cohn et al. (1928). The total Hydrogen ion concentration for varying ionic strengths in the aqueous aerosol is shown for OA = Acetic acid in fig. S2 of the Supplement. For OA = Acetic acid, organic acid dissociation considering $I = \{0.02, 0.2, 1, 4\} \ \mathrm{mol \ kg^{-1}}$ results in approximately $270 - 380\%$ higher $[\mathrm{H^+}]_{\mathrm{tot}}$ than without acid dissociation. Both Acetic acid ($pK_a = 4.76$, Goldberg et al. (2002)) and Decanoic acid ($pK_a = 4.9$, Martell and Smith (1974)) are straight chain monocarboxylic acids with**

 **comparable bulk acidity and their aqueous dissociation properties are expected to be similar. Therefore, variation in $I$ is expected to result in similar $[\mathrm{H^+}]_{\mathrm{tot}}$ in the aqueous aerosol for OA = Decanoic acid as for OA = Acetic acid."**

*Line 289: the sentence containing "pKabulk+1" is confusing. pKabulk+1 does not need an extrapolation to be obtained from pKabulk, but properties at these pHs do.*

100

We use the modified values of the acid constant (not the pH), specifically $pK_a = pK_a^{\mathrm{bulk}} + 1$ and $pK_a = pK_a^{\mathrm{bulk}} + 2$, to represent the surface modulated *response* in dissociation of the organic acid to a given pH of the bulk solution. These shifted $pK_a$ values broadly cover the observations with surface sensitive X-ray Photoelectron Spectroscopy (XPS) of suppressed dissociation states for surface adsorbed propanoic, butyric, pentanoic, octanoic, and decanoic acids across a very wide range of

105 solution pH $= 2 - 12$ (Prisle et al., 2012; Öhrwall et al., 2015b; Werner et al., 2018). We have modified the sentence in the revised manuscript Section 2.4.3 to clarify this:

**"We consider two magnitudes of this apparent shift, covering the range of experimental observations from XPS. For a monoprotic acid, we consider $pK_a = pK_a^{\mathrm{bulk}} + 1$ and $pK_a = pK_a^{\mathrm{bulk}} + 2$, where $pK_a^{\mathrm{bulk}}$ is the well known $pK_a$ of the**

110 **organic acid in aqueous bulk solution."**

*Figures 1-6: all these results are only for pKabulk, pKabulk+1, and pKabulk+2. The pKas of malonic and decanoic acids are different (2.8 and 4.9, respectively). At a certain pH, these 2 acids in an aerosol can have behaviors quite different than described in this paper. In an aerosol with a more realistic composition, more acids have even more pKas. The authors should*

115 *explore the behavior of such acid mixture, or at least, that of individual acids in a wide range of pH, not just near their pKa.*

In line with our response to the previous point, $pK_a = pK_a^{\mathrm{bulk}}$ is the well known acid constant describing its dissociation behaviour with varying pH in a bulk aqueous solution, and $pK_a = pK_a^{\mathrm{bulk}} + 1$ and $pK_a = pK_a^{\mathrm{bulk}} + 2$ are used to describe the experimentally observed surface modulation of the dissociation response to a given pH of the aqueous bulk solution (Werner

120 et al., 2018; Prisle, 2023). We have used the notation $pK_a^{\mathrm{bulk}}$ to differentiate the well known bulk value of the acid constant from the introduced surface modulated suppressed dissociation responses of the acid represented by increasing the bulk acid constant by 1 (moderate suppression of dissociation at a given pH) or 2 (strong suppression of dissociation at a given pH) pH units. These values do not represent the aerosol pH. It is the intrinsic property of the acidic OA we are modifying by assuming these different values for the $pK_a$, according to the observations made with surface sensitive XPS for aqueous solutions of

125 simple monocarboxylic acids over a wide range of pH (Prisle et al., 2012; Öhrwall et al., 2015b; Werner et al., 2018). By shifting the $pK_a$, we are modifying the acid dissociation response to a given pH, which is the cloud pH. We assume cloud pH $= 5$ in all our calculations, consistent with pH of warm low lying tropospheric clouds (Pye et al., 2020). The $pK_a$ of the two organic acids used here to represent OA are quite different ($pK_{a1} = 2.8$ and $pK_{a2} = 5.7$ for malonic acid and $pK_a = 4.9$ for decanoic acid). Their dissociation responses to a given pH are therefore also very different. By using these two organic acids

130    to represent OA, we can assess the significance of surface modulated dissociation response to cloud pH for a range of acidic OA. We have clarified this by revising the description of cloud pH in Methods Section 2.3 in the revised manuscript:

"$[\mathrm{H}^+]_{\mathrm{initial}} = 2.5 \times 10^{-6}\ \mathrm{mol\,L}^{-1}$ **is the Hydrogen ion concentration obtained from the cloud** $\mathrm{pH} = 5$**, which is assumed to be uniform for all size bins and consistent with the** $\mathrm{pH}$ **of warm low lying tropospheric clouds (Pye et al., 2020).**"

135

We have also clarified and expanded the Methods Section 2.4.3 in the revised manuscript:

**"We now introduce a simple empirical representation of the shift in organic acid dissociation previously observed in surface-sensitive XPS experiments. Werner et al. (2018) found that the surface specific dissociation state of surface**
140    **active mono-carboxylic acids was significantly suppressed in dilute aqueous solutions across a very wide range of solution** $\mathrm{pH} = 2-12$**. Similar suppressed dissociation states were also found for other mono- and dicarboxylic acids of both stronger and weaker surface activity, in aqueous solutions closer to neutral** $\mathrm{pH}$ **(Prisle et al., 2012; Werner et al., 2014; Öhrwall et al., 2015a). The shifted dissociation states are attributed to both increased concentrations of the surface active organic acids in the surface and increased non-ideality (higher activity coefficients) of the charged deprotonated**
145    **conjugate species** $\mathrm{A}^-$ **and hydronium ions, compared to the neutral molecular acid** $\mathrm{HA}$**, in the organic-rich air–solution interfacial region (Werner et al., 2018; Prisle, 2023). From eq. 22, this corresponds to an apparent shift of the acid** $pK_a$ **at the surface,**

$$pK_a = pK_a^{\mathrm{bulk}} + \log\left(\frac{\gamma_{\mathrm{H_3O^+}}\gamma_{\mathrm{A}^-}}{\gamma_{\mathrm{HA}}}\right), \tag{1}$$

**compared to the well known bulk acidity** $pK_a^{\mathrm{bulk}}$ **obtained for dilute aqueous solutions, where all activity coefficients**
150    **are assumed to be ideal,** $\gamma_i = 1$ **(Prisle, 2023).**

**The dissociation states observed with XPS are broadly consistent with a magnitude of the apparent shift in** $pK_a$ **of** $\log\left(\frac{\gamma_{\mathrm{H_3O^+}}\gamma_{\mathrm{A}^-}}{\gamma_{\mathrm{HA}}}\right) = 1-2\ \mathrm{pH}$ **units across the surface titration curve (Prisle, 2023). We here introduce the effect of surface modulated acid dissociation by shifting the well known bulk** $pK_a$ **of each organic acid according to these shifts of the surface titration curves. We consider two magnitudes of this apparent shift, covering the range of experimental**
155    **observations from XPS. For a monoprotic acid, we consider** $pK_a = pK_a^{\mathrm{bulk}} + 1$ **and** $pK_a = pK_a^{\mathrm{bulk}} + 2$**, where** $pK_a^{\mathrm{bulk}}$ **is the well known** $pK_a$ **of the organic acid in aqueous bulk solution. To represent the surface shifted dissociation of both carboxylic groups in a diprotic acid, we increase both the first and second acid constant, by 1 or 2** $\mathrm{pH}$ **units, to similarly obtain** $pK_a^{\mathrm{bulk}} + 1$ **and** $pK_a^{\mathrm{bulk}} + 2$**. We here refer to the shifted** $pK_a$ **values as the surface modulated** *apparent* $pK_a$**. However, we strongly emphasize that the** $pK_a$**, which is an intrinsic property of each organic acid in bulk aqueous**
160    **solution, is not itself changed. Only the dissociation responses of the organic acids to a given** $\mathrm{pH}$ **of the solution (here, the cloud** $\mathrm{pH}$**) are changed in the surface (Prisle, 2023). For both mono- and diprotic acids, the values used for surface modulated apparent** $pK_a$ **are given in Table 3. For each** $pK_a$**, the corresponding acid dissociation degree** $\alpha$ **is calculated for the monoprotic acid using eq. 26 and for the diprotic acid using eq. 30. The value for** $\alpha$ **decreases with increasing**

$pK_a$, such that the increased apparent $pK_a$ represent suppressed dissociation of the organic acid in the surface. The surface modulation of organic dissociation is most pronounced in a range of several $\mathrm{pH}$ units around the bulk $pK_a$. At very low and very high $\mathrm{pH}$, the surface dissociation states collapse onto the well known bulk solution dissociation behavior (Werner et al., 2018; Prisle, 2023). For both the organic acids used here, the $pK_a^{\mathrm{bulk}}$ is within a few $\mathrm{pH}$ units of the cloud $\mathrm{pH}$. "

**1.3  Technical corrections**

*Line 215: "F-" and "HF" should be "A-" and "HA", respectively*

The sentence has been corrected in the revised manuscript.

*Many numbers in the manuscript have too many significant digits, e.g., "14.70%", "83.14%", "21.73%", and "64.72%" in Line 403*

We have reduced the number of significant digits to three throughout the revised manuscript.

**2  Reviewer #2**

**2.1  General comments**

*This manuscript focuses on how different chemical properties at aerosol surfaces can potentially modify cloud responses to organic aerosol. It is a detailed look that attempts to bridge from very detailed chemical processes to large scale impacts on climate. The authors are commended for attempting to bridge from the nanoscale to the macroscale, though many assumptions are needed in order to bridge these spatial gaps. The emphasis that this is empirical in the abstract and strong use of caveats in the text is key, as I worry a manuscript like this, even with the caveats, could be over generalized by others later on, but it is still important work. While there are some suggested edits and revisions, I think a revised manuscript should be considered for publication and will provide a unique and interesting contribution to the field.*

Thank you for these encouraging comments. In addition to the edits made according to the suggestions given below, we have also included a concluding Discussion paragraph in Results and discussions Section 3 of the revised manuscript, which addresses the applicability of the presented results beyond the assumptions made in this study. We also refer to our response to the general comments of Reviewer #1.

**2.2 Specific comments**

*It would be helpful in the abstract if it was stated more clearly that what the author are probing is enhanced acidity at the aerosol surface. This can be inferred, but a direct statement would be helpful.*

We are investigating the impacts of two possible dissociation effects of the organic fraction: 1) dissociation according to the well known bulk acidity of organic aerosols in aqueous solutions, and 2) the potential additional impacts of recently observed surface modulated dissociation of surface active acids (Werner et al., 2018; Prisle, 2023). The latter case corresponds to droplets where surface adsorbed acids comprise a significant fractions of the total organic aerosol mass (Gérard et al., 2016; Petters and Petters, 2016; Nozière et al., 2017; Kroflič et al., 2018; Gérard et al., 2019). We have edited the Abstract to reflect this point:

**"The degree of dissociation has recently been observed for several atmospheric surface-active organics with Brönsted acid character to be significantly shifted in the surface, compared to the bulk aqueous solution. In addition to the well known bulk acidity, we therefore introduced an empirical account of this surface modulated dissociation to further explore the potential impact on aerosol climate effects."**

Regarding the reviewer's reference to "enhanced acidity" at the aerosol surface, we want to emphasize that we here deliberately focus on the acid dissociation state at the aqueous surface, which is the observable quantity, rather than making any distinction of a surface specific acidity as a unique property, or assess the underlying mechanisms. We represent the observed change in acid dissociation response at the surface to a given aqueous solution pH by introducing a modulated effective acid constant $pK_a$. We have emphasized this point in the Introduction section of the revised manuscript, in connection with changes made in response to the next comment of the reviewer below.

*A new article came out around the time the authors submitted in PNAS showing for 3 micron aerosol that enhanced acidity can be observed at the surface of particles.Gong et al. (2023) This manuscript bolsters the importance of the authors work and would make sense to include during the revision process.*

Thank you bringing this paper to our attention. The work of Gong et al. (2023) concerns an effect observed exclusively for small droplets (diameter = 2.9 µm) of aqueous 300 mM $NaHSO_4$ mixed with 50 mM $Na_2SO_4$. An enhanced surface propensity of protons was observed in these droplets using stimulated Raman scattering (SRS) microscopy. The effect of surface modulation in organic acid dissociation observed in recent XPS measurements (Prisle et al., 2012; Öhrwall et al., 2015b; Werner et al., 2018) for common atmospheric aerosol components is believed to be primarily linked to the surface activity of the organic acids (Prisle, 2023). The observations were made for macroscopic solutions containing atmospherically relevant organic molecules and is believed to be a general phenomenon for solutions of all dimensions containing surface active acids. It is possible that the new observations of Gong et al. (2023) contribute to the mechanisms behind the previously observed

surface modulation of surface active organic acid dissociation state. Here, we investigate the impacts of organic aerosol acid dissociation in response to droplet bulk pH, defined by the cloud pH in the model, and are concerned with the climate implications of the observed shifted dissociation response, regardless of the underlying mechanism. We have included a discussion of the work of Gong et al. (2023) in the Introduction section of the revised manuscript, in connection with a broader discussion of recent evidence for surface modulation of acid dissociation, compared to behavior described by the well known bulk acid constant $pK_a$:

"Organic aerosols contain a substantial fraction of species exhibiting Brönsted acid character (Jacob, 1986; Millet et al., 2015; Keene and Galloway, 1984; Chebbi and Carlier, 1996; Chen et al., 2021b; Angelis et al., 2012; Mochizuki et al., 2016; Wu et al., 2020; Kawamura et al., 1985). The concentrations of acidic species in aqueous aerosols directly affect the aerosol pH by modifying the $H^+$ concentrations within the aerosol (Pye et al., 2020; Ault, 2020). This pH affects the dissociation of individual acidic species with significant consequences for aerosol chemistry (Hung et al., 2018; Wang et al., 2018) and phase state (Liu et al., 2019). For example, pH dependent Sulfur oxidation (Liu et al., 2020) and salt formation by acidic or basic OA (Yli-Juuti et al., 2013) can each lead to significant mass formation and alter the overall chemical composition of aerosols. The chemical form (protonated or deprotonated) of acidic OA and contributions to the number of solute species in the aqueous aerosol phase can strongly affect water activity and condensation–evaporation equilibrium (Prisle, 2006; Prisle et al., 2008; Frosch et al., 2011; Michailoudi et al., 2019)."

and further:

"The general behavior of acidic compounds at the aqueous interface is still not well constrained (Saykally, 2013). Petersen and Saykally (2005, 2008) observed an enhanced surface concentration of hydronium ions in aqueous solutions of Hydroiodic acid, alkali iodides and alkali hydroxides using second harmonic generation spectroscopy experiments. This was in contrast to previous macroscopic bubble and droplet experiments, which were interpreted to indicate that hydroxide ions were enhanced at the air–water interface (Graciaa et al., 1995; Takahashi, 2005; Karraker and Radke, 2002; Creux et al., 2007). Enami et al. (2010) also made similar observations of enhanced hydronium ions in the surface of Trimethylamine solutions using electrospray mass spectrometry. Recently, Gong et al. (2023) used stimulated Raman scattering microscopy to observe enhanced concentrations of Sulfate and Bisulfate anions, with Bisulfate being more surface enriched than Sulfate, in the surface of $2.9$ μm aerosol droplets generated from an aqueous solution with $300$ mM $NaHSO_4$ and $50$ mM $Na_2SO_4$ at the same pH. They interpret this as an enhancement of acidity, with approximately threefold increase in the Hydrogen ion concentration, at the droplet edge, compared to the center of the droplet. Previous observations by Margarella et al. (2013) on the dissociation of Sulfuric acid at the water interface using liquid-jet photoelectron spectroscopy, have also reported that the ratio of Bisulfate-to-Sulfate anions was higher in the surface region."

*My one major gripe is that ionic strength is really important for these types of aerosols and is not mentioned at all. This is not meant to single out the authors here, as a number of other manuscripts recently in this space also don't discuss it even when submicron particles (and particularly ultrafine particles) will have very very high ionic strengths. Activity coefficients are mentioned briefly, but the connection to ionic strength and the implications for this work are really key. In a low water environment, which most of the submicron particles in this model will be high ionic strengths leading to activity coefficients that are « 1 (or bouncing back to » 1 for really high ionic strengths) are central to modified acid dissociation behavior. While the discussion kind of talks around it, the manuscript would be significantly improved if it directly discussed ionic strengths and their importance under the non-dilute conditions of aerosols.*

We agree that ionic strength is important for the types of aerosols discussed here. As explained in a previous response to Reviewer #1, we have used an ionic strength of $0.5 \, \mathrm{mol \, kg^{-1}}$ in our calculations, because this is the condition for which the modified sulfur chemistry mechanism was defined for all organic acids (Liu et al., 2020). We realised that this condition and reason for choosing it were not explicitly mentioned in the manuscript and thank both the reviewers for bring this up. We have now added an explanation for the choice of ionic strength in the Methods Section 2.3 of the revised manuscript. Furthermore, we note that ionic strength is a bulk phenomenon and not expected to affect surface adsorbed organic acids, which are partially liquid-liquid phase separated, at least not to the same degree as solutes in the droplet bulk.

The sulfur chemistry rate constants are to our knowledge not available for varying ionic strengths and we were therefore not able to show or discuss with certainty the secondary sulfate production or the cloud activation processes for varying ionic strengths. However, the measured $pK_a$ for aqueous malonic acid bulk solutions of varying ionic strength was available from Liu et al. (2020). Since the calculation of total hydrogen ion concentration in the aqueous aerosol does not require any additional rate constants, we were able to calculate $[\mathrm{H^+}]_{\mathrm{tot}}$ for varying ionic strength considering bulk ($pK_a = pK_a^{\mathrm{bulk}}$) and surface modulated ($pK_a = pK_a^{\mathrm{bulk}} + 1$ and $+2$) organic dissociation. We have added new figures (fig. S1 and S2) showing results of these calculations in the Supplement Section S2, and a discussion in the Results and discussion Section 3.1 of the revised manuscript. Further details on the calculations are provided in our response to first specific comment from Reviewer #1.

*Line 93: The word "the" is spelled "thhe"*
The sentence has been corrected.

*For the simulation in Table 1, the total number concentration is quite low (700 $\#/cm^3$). A little more justification for why such a low concentration was used would be helpful.*
The total aerosol number concentration given in Table 1 is representative for clean environments, such as European villages, obtained from measured size distributions reported in Tunved et al. (2005, 2008). We have added the reference in the table as well as modified the description in the revised manuscript:

**"The initial number concentration in each sub-range (Table 1) used for all HAMBOX simulations is representative of clean environments, such as European villages (Tunved et al., 2005, 2008)."**

*Perhaps I missed it, but I didn't see any discussion on organosulfur compounds and their impacts on aerosol sulfur. I think a few lines on the extent of its formation would be helpful.Brüggemann et al. (2020); Riva et al. (2019); Chen et al. (2021a); Hettiyadura et al. (2019)*

Thank you for this suggestion. We agree that the increased sulfate concentrations in the aerosol discussed in this work could also impact the organosulfur formation in the atmosphere. Furthermore, many organosulfur compounds are both surface active and acidic, and could therefore be expected to display similar surface modulated dissociation behavior in aqueous droplets as the carboxylic acids discussed here. We have added a discussion on this in Results and discussions Section 3.6 of the revised manuscript:

**"The increased Hydrogen ion concentration in aqueous aerosols as a result of OA acid dissociation leads to enhanced Sulfate mass from oxidation of $SO_2$ by $H_2O_2$. Increased Sulfate concentrations could potentially lead to enhanced formation of organosulfur compounds within the aerosols. Organosulfur compounds form in the atmosphere through heterogeneous reactions between volatile organic compounds and inorganic aerosol Sulfate and can comprise over 15% of the secondary organic aerosol mass (Brüggemann et al., 2020; Riva et al., 2019; Chen et al., 2021a; Hettiyadura et al., 2019). These organosulfur compounds could further increase OA mass and affect the resulting cloud droplet number concentrations. Organosulfates have been shown to exhibit acidic properties and primarily exist in the deprotonated form under atmospheric $pH$ conditions (Fankhauser et al., 2022). Organosulfates are also known to be surface active in aqueous solutions (Hansen et al., 2015; Prisle et al., 2010b, 2011; Lin et al., 2018; Malila and Prisle, 2018). Therefore, the mechanisms of both organic acid dissociation and its surface modultation studied here for atmospheric carboxylic acids could also apply to organosulfate aerosols, potentially affecting cloud droplet activation in a similar manner."**

*In the direction of organosulfates their impact at the surface, particularly given recent working showing that they are primarily deprotonated,Fankhauser et al. (2022) and potentially surface active would be worth briefly discussing, even if not accounted for in the already extensive calculations and results.*

This is an excellent suggestion. We have added a comment on this in Results and discussion Section 3.6 of the revised manuscript, as given in the previous response.

*Another topic that I think could impact these results is aerosol mixing state. The work of Nicole Riemer showing impacts on CCN for externally versus internally mixed populations is worth noting.Ching et al. (2016); Razafindrambinina et al. (2023) As is the definition in her review from 2019 of "physicochemical mixing state" to describe not only compositional differences*

*particle-to-particle, but also within individual particles (Figure 3).Riemer et al. (2019)*

The results of our calculations will be impacted by the aerosol mixing state. We use the SALSA2.0 aerosol module developed by Kokkola et al. (2018) which considers internal mixing of three species (sulfate, organic, and sea salt), and external mixing with two other species (black carbon, mineral dust). Internal mixing is considered for the highly water-soluble species and external mixing for the insoluble species. This mixing state is considered as a realistic representation of atmospheric aerosol relevant for cloud activation processes (Kokkola et al., 2018). We have added a brief description of the mixing states used in the present work in Methods Section 2.1 of the revised manuscript.

**"Of these model compounds, Sulfate, Organic aerosol, and Sea salt constitute the soluble species and are considered as internally mixed in each size bin of the aerosol population. Black carbon and Mineral dust are insoluble species which are externally mixed in each size bin with the soluble species, as described by Kokkola et al. (2018)."**

*Line 155 it says they follow the procedure from Liu et al. 2020, which is valid for $\mathrm{pH} > 2$, but what if it is $< 2$, which is common in the atmosphere (per Pye et al., which they cite).*

For our current work, the cloud $\mathrm{pH}$ is taken as the droplet solution pH. Current large-scale models (Sect. 8, Pye et al. (2020)) and observations (Sect. 7, Pye et al. (2020)) indicate that cloud $\mathrm{pH}$ has a global mean somewhere between $4$ and $6$ and ranges from around $2$ to above $7$ (Fig. 2, Pye et al. (2020)). We have considered warm low level clouds in our simulations and for these types of clouds, $\mathrm{pH} > 2$ seems reasonable. We have added this explanation in Methods Section 2.3 of the revised manuscript:

**"$[\mathrm{H^+}]_{\mathrm{initial}} = 2.5 \times 10^{-6} \ \mathrm{mol \, L^{-1}}$ is the Hydrogen ion concentration obtained from the cloud $\mathrm{pH} = 5$, which is assumed to be uniform for all size bins and consistent with the $\mathrm{pH}$ of warm low lying tropospheric clouds (Pye et al., 2020)."**

The choice of the boundary condition $\mathrm{pH} > 2$ is also considered reasonable for investigating specifically the effects of surface modulated organic dissociation. At very low or high bulk $\mathrm{pH}$ values, the surface modulation of $pK_a$ was observed to collapse onto the well-known bulk solution dissociation behavior (Werner et al., 2018; Prisle, 2023). Therefore, focusing on $\mathrm{pH}$ values above 2 allows for a more distinct examination of the influence of surface-modulated $pK_a$ on cloud properties, as deviations from the bulk behavior are most pronounced within this $\mathrm{pH}$ range. We have added this point in Section 2.4.3 of the revised manuscript:

**"The surface modulation of organic dissociation is most pronounced in a range of several $\mathrm{pH}$ units around the bulk $pK_a$. At very low and very high $\mathrm{pH}$, the surface dissociation states collapse onto the well known bulk solution dissociation behavior (Werner et al., 2018; Prisle, 2023). For both the organic acids used here, the $pK_a^{\mathrm{bulk}}$ is within a few $\mathrm{pH}$ units of the cloud $\mathrm{pH}$."**

*Line 217 a space between the two l's in "moll-1"*

This has been corrected in the revised manuscript.

370 *A different topic worth mentioning, particularly around line 265, even if it is not the focus of this work, is phase (solid, semi-solid, vs. liquid) and liquid liquid phase separations, which could provide a different interface and complicated enhanced inter-facial active region. Reviews by Reid et al. (2018), Freedman (2017) and You et al. (2014) could be worth mentioning or references therein.*

375 Bulk-surface partitioning can be considered as a form of (second order) liquid-liquid phase separation. The suppression of organic acid dissociation is exactly a consequence of the increased concentrations in the surface phase (Prisle, 2023). In our calculations, we make the simplifying assumption that the droplets are well mixed and instead consider the impact of surface partitioning on the organic dissociation behavior for the droplet as a whole, without explicitly accounting for the distribution of components at the droplet surface. Instead, the overall dissociation property in the entire droplet is adjusted to represent the

380 effects of surface partitioning. We explain this in Methods Section 2.4.3 in the revised manuscript:

**"Although we implement the effect of surface modulated acid dissociation as a consequence of simultaneous surface activity of the organic acid, we do not explicitly consider the bulk–surface partitioning of organic acids in our calculations. Our simple empirical representation by shifting the apparent $pK_a$ for the organic acid corresponds to assuming**

385 **that the overall dissociation state in aqueous aerosols and droplets is described by the surface modulated properties. This is closely representative of aerosols and droplets where the majority of organic aerosol components are partitioned to the aqueous surface, as a consequence of strong organic surface activity or high $A/V$ in the microscopic and submicron size ranges (Prisle, 2021, 2023). In real atmospheric aerosol and droplet mixtures of both surface active and more water soluble OA, organic species will be partially partitioned to the surface and the overall dissociation state should be**

390 **described as a combination of both well known bulk acidity and surface modulated states. The present simple empirical representation therefore gives an upper bound of the potential effects of surface modulated acid dissociation according to the previous observations from XPS experiments."**

We have also added a discussion on similar effects for liquid-liquid phase separation in the discussion in Results and dis-

395 cussions Section 3.6 in the revised manuscript:

**"Bulk-surface partitioning in aqueous aerosols can be seen as a form of (potentially second order) liquid-liquid phase separation (Prisle, 2023). Phase separation of organic aerosols and its impact on cloud activating properties of aerosol particles have been widely studied (Reid et al., 2018; Freedman, 2017; You et al., 2014). The partitioning of surface**

400 **active aerosol components occurs between the bulk and surface phases due to differences in composition and affinity**

**for each phase. The suppression of organic acid dissociation considered here is exactly a consequence of the increased concentration of surface active organic acid in the surface phase. Liquid-liquid phase separation in the bulk phase would effectively create two separate solutions with different compositions and ensuing properties. The modulation of organic acid dissociation could be taken into account separately for these phases, based on their individual concentra-**
405   **tions, following analogous schemes as described by Prisle et al. (2010a)."**

*Perhaps I missed this, but what depth is assumed for this modified dissociation? I think this should be explicitly state somewhere.*

Our empirical treatment is based on the approximation that, due to bulk–surface partitioning, the majority of the organic
410   exists in the surface, which has been shown to closely correspond to the droplet state for many finite sized systems (Prisle et al., 2010b, 2011; Prisle, 2021). In small droplets, the surface-to-bulk volume ratio is large enough that the whole droplet may be represented by surface properties. Please also refer to changes made in response to the previous and the following comment. In addition, we have added the following to the Introduction:

415   **"In microscopic and submicron-sized aerosols and droplets, the surface adsorption can result in significant redistribution of surface active OA mass from the bulk to the surface phase, so-called bulk–surface partitioning, as a consequence of the very high surface area ($A$) to bulk volume ($V$) ratio in these size ranges (Prisle et al., 2008, 2010b). For spherical droplets of diameter $D_{\mathrm{wet}} = 0.1$, 1, and 10 µm, $A/V = 6/D_{\mathrm{wet}}$ is 60, 6, and 0.6 µm$^{-1}$, respectively (Prisle, 2021). Thermodynamic calculations have shown that for aerosol particles containing surfactant fatty acids and their salts,**
420   **organosulfates, di- and polycarboxylic acids, and complex fulvic acids, a large fraction of the surface active OA is partitioned to the surface during major parts of hygroscopic growth and cloud droplet activation (Prisle et al., 2010b, 2011; Hansen et al., 2015; Malila and Prisle, 2018; Lin et al., 2018, 2020; Prisle, 2021; Vepsäläinen et al., 2022, 2023). Consequently, the chemical and physical state of the surface may significantly contribute to determining the overall aerosol properties (Prisle et al., 2012; Bzdek et al., 2020; Prisle, 2021, 2023)."**

425

*Also, how long do the molecules spend in that volume? Is the reaction so fast near the surface that the fact most of the particle has a more "normal" pKa, in other words one that is not the modified dissociation at the surface, does not matter?*

Properties corresponding to surface adsorbed organics are expected to be representative throughout simulations. The simu-
430   lations run for 1 hour and the bulk and surface are assumed to be within reasonable proximity of equilibrium conditions after approximately 495 seconds (Noziere et al., 2014). Lin et al. (2020) investigated the impact of surface adsorption dynamics on surfactant effects in cloud droplet activation and found that impacts different dynamic effects nearly cancel out at every time step. Therefore, we consider use of equilibrium conditions for these simulations a good first approximation. It is possible that this is not the case for all types of surfactant aerosols, but currently dynamic surface tension data is not available to perform a
435   similar analysis for any other systems than those investigated by Lin et al. (2020). Therefore, consideration of such dynamic or

kinetic effects is beyond the scope of the present simulations. We have clarified this point in the Methods Section 2.4.3 in the revised manuscript.

**"When surface modulated organic acid dissociation is considered, these properties are assumed to remain consistent throughout the 1-hour simulations. Prisle et al. (2008) and Prisle (2021) estimated that surface adsorption of typical atmospheric surfactants equilibrate within a timescale of a second in micron-sized droplets. Lin et al. (2020) investigated the impact of surface adsorption dynamics on surfactant effects in cloud droplet activation and found that different dynamic effects nearly cancel out at every time step. Noziere et al. (2014) assumed that both the bulk and surface reach a state of reasonable equilibrium with respect to organic adsorption at the aqueous surface within approximately 495 seconds. Therefore, we consider this assumption to be a reasonable first approximation."**

*Line 353, the first sentence of this paragraph is missing a word or confusingly written, please revise.*
The sentence has been corrected in the revised manuscript.

*While the authors lay out in the text what the x-axis is in Figure 1, 2, 4. It is still non-intuitive to lots of readers. It would be helpful if the authors could either add the explanation again to each caption or annotate the figure further to help clarify what that axis means (I had to re-read a number of times, and I was reading pretty intently). The figures with the chi as organic fraction are more intuitive.*

We agree that the x-axis labels may be difficult to follow and this was discussed extensively between the authors for the previous version of the manuscript. Following reviewer's comment, we have reverted to an earlier version of the x-axis labels, which we hope are more intuitive. Concrete suggestions for further improvements are also most welcome.

*Figure A3 caption, "od" should be "of"*
The figure caption has been corrected in the revised manuscript.